# Integrating psychosocial health into disaster risk management: Insights from COVID-19 in Durán, Ecuador

Mercy J. Borbor-Cordova[1,2]*, Madison Searles[3], Heydi Roa-López[2,4],
María del Pilar Cornejo-Rodriguez[1,2], Katty Castillo[2], Andrea Orellana-Manzano[5],
Christina D. Campagna[3]*

1 Faculty of Maritime Engineering and Marine Sciences, Escuela Superior Politécnica del Litoral, ESPOL, Guayaquil, Ecuador, 2 Pacific International Center for Disaster Risk Reduction (PIC-DRR), ESPOL, Guayaquil, Ecuador, 3 Global Health Institute, State University of New York Upstate Medical University, Syracuse, New York, United States of America, 4 Faculty of Natural Sciences and Mathematics, Escuela Superior Politécnica del Litoral, ESPOL, Guayaquil, Ecuador, 5 Laboratorio de Farmacología Molecular Aplicada, Facultad de Ciencias de la Vida (FCV), Escuela Superior Politécnica del Litoral, ESPOL, Guayaquil , Ecuador

* meborbor@espol.edu.ec (MBC), campagch@upstate.edu (CC)

## Abstract

The COVID-19 pandemic and intensifying climate-related disasters have highlighted the necessity of integrating psychosocial health and disaster risk management strategies. In the post-pandemic context, it is important to understand the psychosocial vulnerability factors that exacerbate stress and depression in order to inform interventions that enhance resilience during disaster preparedness and recovery efforts. Self-reported stress and depression were evaluated to assess risk and protective factors using questionnaires that were completed by 340 out of 446 participants in Durán, Ecuador in 2021. Considering social vulnerability factors of exposure, sensitivity and adaptive capacity at individual and urban levels, this study applied Kendall's tau and odds ratio analyses to explore associations between stress and depression, identifying risks and protective factors within the sample population. Women (68.5%) self-reported to be vulnerable to moderate stress (67.8%) and mild depression (29.6%). There was a positive association between stress and depression levels (0.42, *p: < 0.02*). Odds ratio indicated adults experienced moderately severe stress, while those who were in domestic partnership had a 20% reduced risk of stress. Participants living in collective housing and rooms in tenement housing and apartments were more likely to experience higher levels of stress. Depression levels are associated with chronic illnesses (p < 0.02), residing in areas lacking paved roads *(p < 0.005)*, without access to water *(p < 0.036)*, and without COVID-19 vaccination *(p < 0.043)*. The Duran study revealed the complex nature of psychosocial health, shaped by individual vulnerabilities, deficient urban infrastructure, and limited capacities of city systems. In the long-term recovery process, the integration of mental health and disaster risk management (DRM) would focus on community-engagement

**Data availability statement:** All relevant data are available within the paper and its Supporting Information files, in compliance with Ecuador's Ley Orgánica de Protección de Datos Personales and national data protection regulations. DOI: 10.6084/m9.figshare.28784126 Link: https://figshare.com/articles/dataset/Dur_n_2021_COVID-19_Stress_and_Depression_Survey_Dataset/28784126?file=53625053.

**Funding:** Research reported in this publication was supported by the Pacific International Center for Disaster Risk Reduction at ESPOL (CIPRRD-03-2021), Secretary Risk Management of Ecuador, and the support on the field campaign by the Municipality of Duran.

**Competing interests:** The authors have declared that no competing interests exist.

for preparedness, equitable access to health/mental services, inclusive urban planning, and partnerships intervention to enhance community adaptive capacity to improve resilience and promote sustainable recovery. In line with the Sendai Framework for disaster risk reduction (DRR), humanitarian emergencies, climate and health crises, and compound risks offer critical opportunities to advance resilience in urban and health systems by addressing structural vulnerabilities, enhancing interinstitutional coordination, and integrating mental and psychosocial health into disaster risk reduction and management strategies.

## Introduction

The COVID-19 pandemic, officially declared over in May 2023, left behind a critical lesson: the integration of psychosocial health into disaster risk management (DRM) is essential to build healthy, resilient cities [1,2]. As climate-related disasters and global health crises like COVID-19 continue to expose and intensify systemic vulnerabilities, addressing the psychosocial dimensions of disaster preparedness, response, and recovery has become a cornerstone for building resilient communities [3]. The pandemic disrupted economies, education, and social stability, exacerbating mental health challenges, especially in the cities of the Global South [4,5]. Addressing these issues requires a holistic, intersectoral, and interdisciplinary approach that strengthens resilience at the local level [6,7]. However, there is a gap in empirical research on DRM and psychosocial health in the Global South, limiting the diversity of perspectives and contextualized insights that could contribute to a more comprehensive understanding of effective strategies and best practices across different socio-cultural and economic settings [8].

Psychosocial health encompasses mental, emotional, social, and spiritual well-being and is deeply interconnected with mental health. Large-scale disasters, such as the COVID-19 pandemic, have exacerbated mental health challenges, exposing pre-existing vulnerabilities [9–12]. Social isolation, financial insecurity, misinformation, health, and mental health risks, have been further complicated by chronic neglect and underinvestment in mental health care worldwide [13–16]. Many studies indicate elevated levels of depressive symptoms, anxiety, stress, and other mental health conditions across diverse populations, especially in vulnerable populations and disadvantaged groups across regional and cultural contexts [4,7,9,11,17–23].

Beyond COVID-19, overlapping threads such as climate change, extreme events, pollution, displacement and emerging epidemics interact to generate compound risks that strain health systems, livelihoods, and overall well-being [14,24–26]. Floods, droughts, wildfires, heat waves, and climate-sensitive diseases complicated pandemic surveillance and response, while disruptions to food and water systems and supply chains exacerbated food insecurity, malnutrition, displacement, and psychosocial stress [27–29]. These cascading impacts illustrate the systemic risks highlighted in the Sendai Framework for Disaster Risk Reduction (2015–2030), which accentuates the need to integrate psychosocial and mental health considerations into disaster preparedness, response, and recovery strategies [1,8,17,30]. Aligned to the

Sendai Framework, the Health Emergency and Disaster Risk Management Framework (HEDRM) offers a comprehensive approach to address both physical and mental health risks in emergencies and crises [24,31,32]. Within this systemic risk framework, a deep understanding of the social vulnerability factors that affect psychosocial wellbeing during health crises and multiple hazards should be part of any post-disaster recovery process [2,8,24,29,32–36].The World Health Organization (WHO) HEDRM framework provides a pathway for "Building Back Better" by strengthening resilient health systems and communities through inclusiveness and equity, while reducing inequalities in the different stages of preparedness [24,37–39]. However, epidemics and pandemics remain largely unintegrated into local level, multi-hazard, and intersectoral intervention strategies. [40–43]. Health inequalities are part of these complex interactions that increase the vulnerability of some groups during the pandemic and climate events [8,29,32,35].

Ecuador was significantly impacted by the COVID-19 pandemic, reporting 1 million cases and 36,019 deaths by May 2023 [31]. The health crisis caused job losses, asset depletion, and highlighted the weaknesses in Ecuador's public health system during the pandemic's first year [44]. Studies on psychosocial and mental health in Ecuador revealed that 20.3% experienced moderate to severe depression and 22.5% moderate to severe anxiety, particularly among women and coastal residents [45]. Adolescents (n = 301) increased depression, anxiety, and stress post-confinement [46]. A study in Galapagos (n = 369) revealed 52% of respondents experienced high levels of stress, and women reported higher levels of depression and stress [47]. Risk factors were financial distress, interpersonal conflicts, feelings of isolation, and fear of contagion of COVID-19 [47]. Ecuador is highly exposed to multiple hazards, including climate events, such floods, landslides, and El Nino geologic hazards such as earthquake and volcanoes and biological hazards such as epidemics [48–50]. National findings align with international consensus that vulnerability to disasters is shaped by social, economic, and built environment [51–54]. Recognizing and addressing these vulnerabilities and health inequalities is crucial when designing effective strategies for psychosocial health recovery and resilient cities in the Global South [51–56].

Durán City has been the focus of multiple studies addressing climate-related hazards, including flooding, urban heat islands, landslides, urban vulnerability, as well as COVID-19 vaccination perception, and community participatory research [57–62]. These efforts position the city as an urban laboratory for examining the complex interactions among climate risks, pandemics and social vulnerability in intermediate Latin America cities. Considering the paucity of original research and context-specific evidence, this study offers insights into how vulnerability factors interact during crises, informing strategies to strengthen urban resilience [29,34]. With the onset of the COVID-19 pandemic, DRM initiatives expanded to include health crises, recognizing the compounded risks posed by overlapping climate events and public health hazards [39].

This study aims to assess the levels of self-reported stress and depression among volunteers, recognizing vulnerability and risk factors associated with psychosocial health during the COVID-19 pandemic. The goal is to identify strategies for integrating DRM and mental health support during and post-disaster. This research is part of a larger cross-sectional COVID-19 project exploring enabling factors for developing multihazards early warning systems at city level.

This study considered the following research questions: 1) What are the key risk factors for stress and depression among the general urban population in Durán? 2) Which population groups are most vulnerable to higher levels of self-reported stress and depression? 3) What recommendations can be drawn to inform future post-disaster recovery efforts? Durán's study considered the following hypotheses: a) Individuals with pre-existing conditions or chronic illnesses are more susceptible to stress and depression. b) Urban vulnerability factors—related to the social and built environment—exacerbate stress and depression in vulnerable groups during disasters and health crises. c) Adaptive capacity factors, such as higher vaccination rates, can be associated with lower levels of community vulnerability.

## Methods

### Study area

Durán, an intermediate city in Ecuador with a population of 303,910 is strategically located just 10 minutes from Ecuador's most important economic hub, Guayaquil [63]. Durán functions as a satellite city that provides housing and employment

while attracting manufacturing, industries, and businesses, contributing to its own economic development [64]. Its urban development has expanded across coastal lowlands, mangrove areas, and hills, with the city surrounded by the Guayas Estuary. Limited urban planning has led to the emergence of informal settlements (40% of the city) that often lack secure land tenure, basic services, essential infrastructure, increasing urban security and social vulnerability [59,65].

## Study design and settings

This observational population study was conducted in Durán, Ecuador, during September and October 2021. A non-probabilistic convenience sampling approach was employed to collect serological samples for SARS-CoV-2 antibody testing, indicative of prior infection or vaccination, and to administer a structured survey. The non-probabilistic method used in Durán's study was based on the representativeness of the system (Durán city), by selecting sites based on in-depth knowledge of Durán's urban structure and social vulnerability was essential in mitigating bias during survey administration [59,66]. In doing so, we carefully considered the selection, availability, and quality of the variables applied in the survey, recommended for observational population studies [66].

Data collection was conducted across various representative sites within the city to ensure a comprehensive understanding of the population's psychosocial health in the context of the COVID-19 pandemic. Five strategic sites across the city, including parks, sports courts, and community centers, were selected to recruit based on logistical feasibility, infrastructure adequacy, and safety considerations. These locations were identified to capture diverse population segments and neighborhoods, informed by prior vulnerability assessments related to climate hazards in Durán. Thus, sample size was determined based on the operational capacity of the field campaigns, consistent with the non-probabilistic convenience sampling strategy employed. See Fig 1 for description of the study site and sampling locations [59].

Out of 446 individuals that participated at the study, 106 did not complete either the Perceived Stress Scale (PSS), the Depression questionnaires (PHQ-9), or both, therefore they were excluded from the analysis. The 340 participants that completed the questionnaires met the inclusion criteria of being over 12 years old. The exclusion criteria in this study were applied to individuals aged 12 or younger and those with physical or mental disabilities.

## Survey

A structured survey was administered to participants by ten trained professionals in a face-to-face setting using cell phones or tablets [67]. Participants were assigned a code during the registration process that was used as the identifier for survey data. Written informed consent was obtained from all participants during the field sampling and survey process. For those participants who completed the survey via an App, consent was similarly obtained and documented electronically. Participants were fully informed on the nature and purpose of the study and consented before data collection. Survey responses were sent to an anonymized database for subsequent quality control and data pre-processing. The fieldwork process comprised the following steps: 1) Participant Registration and 2) Informed Consent signed to participate, 3) Application was utilized to deliver the results of the serological test, 4) survey administration; participants completed a structured survey, 5) blood test was applicated, 6) volunteers were notified of the antibodies test by the App or e-mail. It was clearly stated that the interpretation of these results should be conducted by a qualified healthcare provider to ensure accurate understanding and appropriate follow-up. Fig 2 describes the process during the field campaign to recruit volunteers for a serological test and to complete the study survey.

## Vulnerability framework variables

In this study, we apply a vulnerability framework for understanding communities and individuals who are affected by compound hazards such as climate hazards and COVID-19. The framework includes three key dimensions: exposure, sensitivity, and adaptive capacity, and is based on the vulnerability concept used in the climate change and disaster risk reduction community [68,69]. Exposure is the degree which people or systems come into contact with hazards, *sensitivity*

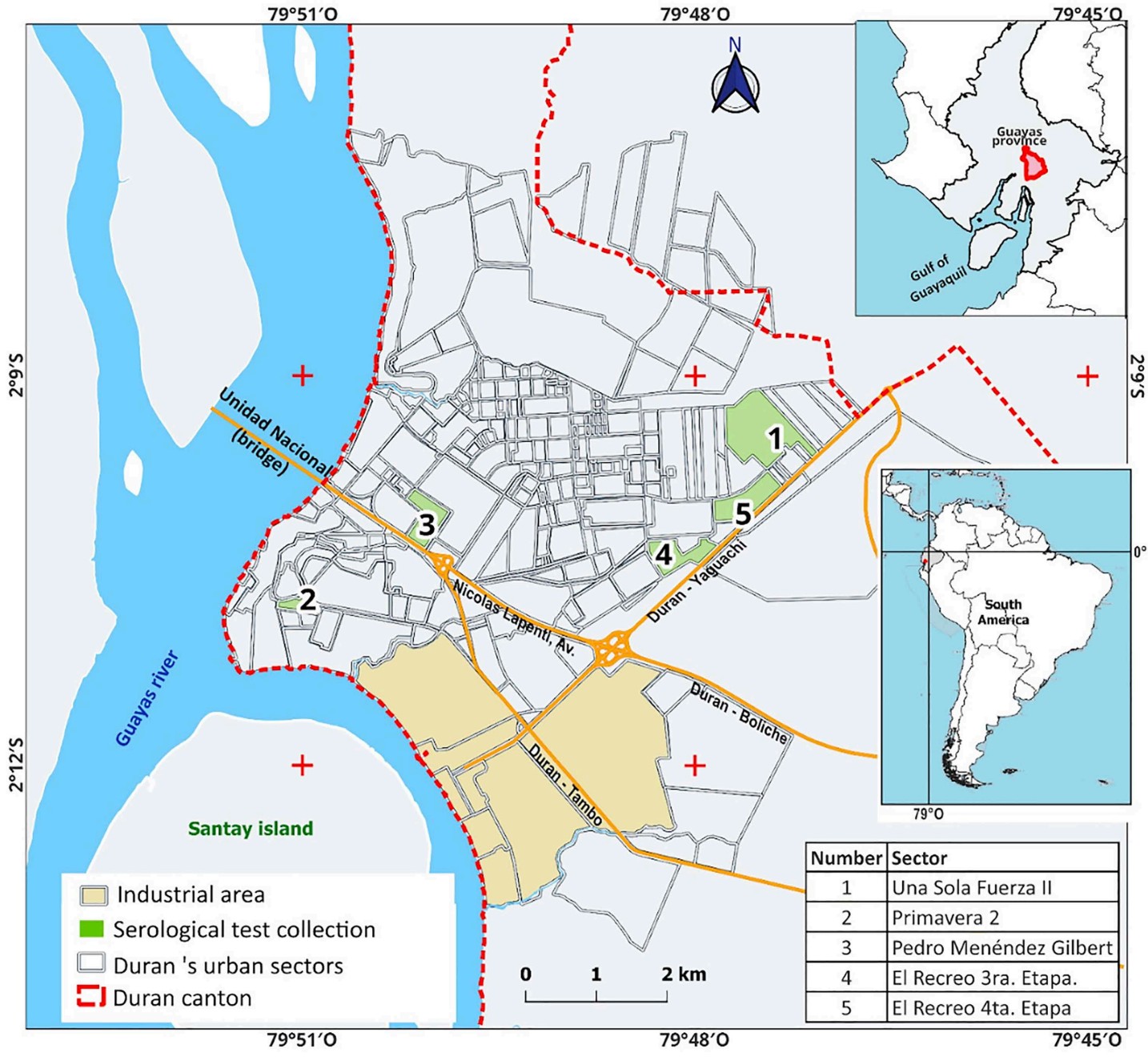

**Fig 1. Location of Durán as a satellite city of Guayaquil, and the 5 sites where the blood tests and surveys were applied.** Sites were selected to capture urban neighborhoods diversity and logistic feasibility.

pertains to factors making individuals or communities more susceptible to negative impacts of hazards, while *adaptive capacity* refers to an individual or community's ability to cope and adapt to stressors before, during, or after a health emergency or disaster [70]. Individual sensitivity includes factors such as sex, age, marital status, comorbidities, drugs/alcohol consumption. Factors of urban sensitivity are related to housing types, access to basic services, and road infrastructure.

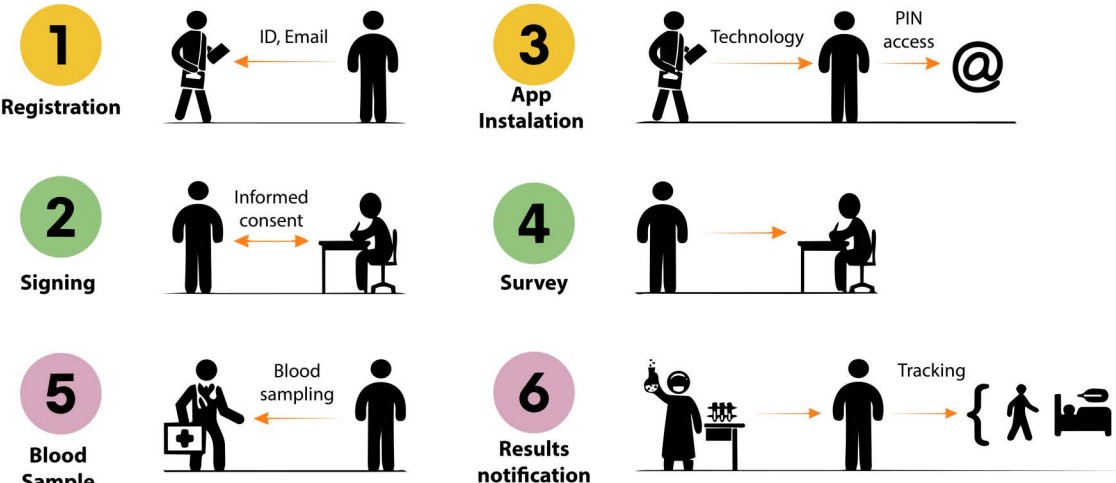

**Fig 2. Diagram of the process from registration (1), informed consent (2), app installation to monitor risks (3), survey (4), and serologic samples for COVID-19 test (5), and results notification (6) during the field campaign in Durán Study.**

The variables associated with adaptive capacity are educational level, occupational status, and COVID-19 immunization. These variables are important in considering one's ability to respond to and recover from crises [6,18]. Exposure includes travel history that may have involved contact with individuals (or themselves) presumed or diagnosed with COVID-19 by the antigen test. See Fig 3.

### Measures

**Perceived Stress Scale (Based on PSS-10 items).**  A modified Spanish version of the Perceived Stress Scale (PSS) was used in the present study and was based on 6 questions from the 10-item self-reported questionnaire and the 4-item questionnaire that assess the level of perceived stress in the last month [71,72]. This study included questions 3, 4, 5, 6, 9 and 10 from the 10-item questionnaire (See supplementary material) Each item is scored on a five-point Likert scale from 0 to 4, except items 4 and 5 which are reverse scored. For the scores given in the present study the internal consistency was (Cronbach's $\alpha = 0.667$), similar to other studies considered adequate psychometric property for reliability in Ecuador [47]. The levels of perceived stress were classified as follows, scoring 0–8 as low perceived stress, 9–16 as moderate perceived stress, and 17–24 as high perceived stress. Previous studies in Ecuador have assessed the PSS-10 and PSS-4 and have recommended the PSS-4 over the PSS-14 for use in Ecuador [73].

**Depression (Based on PHQ-9 questionnaire).**  Patient Health Questionnaire-9 (PHQ-9) is a brief self-report questionnaire assessing the severity of depression symptoms in an adult through nine items [74]. In the present study we used an adapted Spanish version of the PHQ-9 that shows psychometric properties comparable to the original version [75]. Item 9 was not included (suicidal screening) because of cultural sensitivity in the context of COVID-19, despite this, the internal consistency of the measure in this study was high (Cronbach's $\alpha = 0.87$), like previous studies in Ecuador [45].

According to scores given by participants, the symptoms can be classified in five levels: a score of between 1 and 4 points indicates the presence of minimal symptomatology, between 5 and 9 mild, 10–14 moderate, 15–19 moderately severe, and 20–24 severe). The PHQ-9 instrument has been assessed in Low- and Middle-Income countries (LMIC) with recommendations regarding comorbidity factors, survey administration protocols, and language, that were considered in our research [76]. Recent assessments of the PHQ-9 suggest that item 9 may not be adequate to identify individuals at

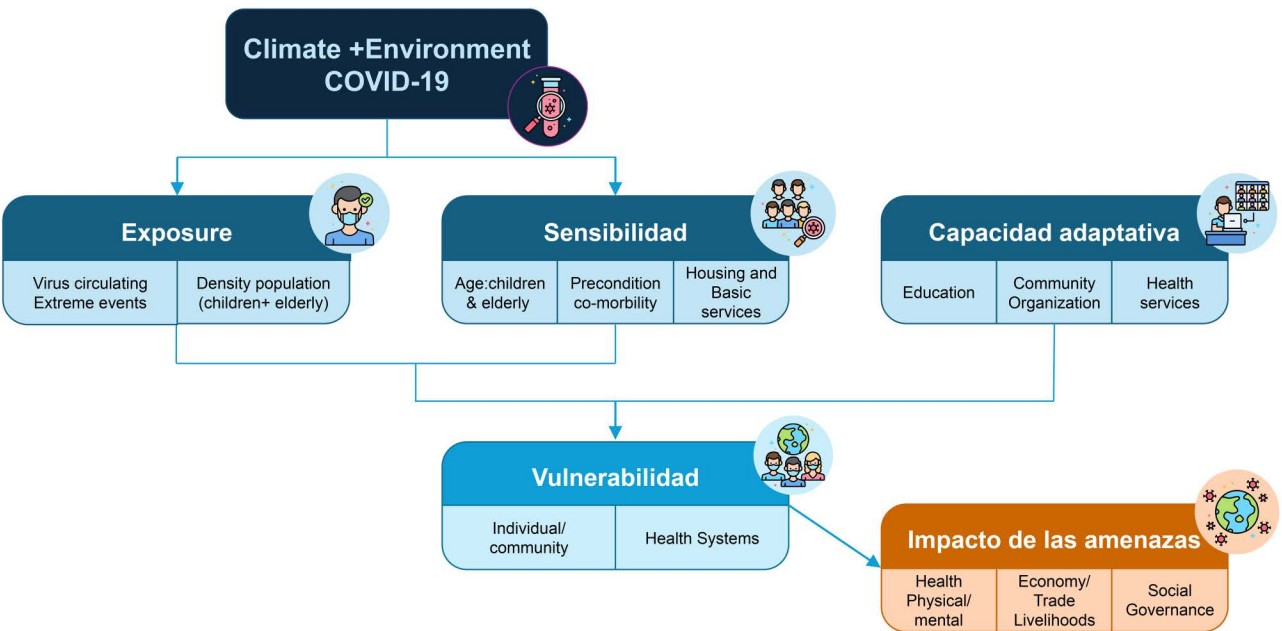

**Fig 3. Diagram of the multi-hazard vulnerability framework, should consider multiple dimensions.** Exposure refers to the degree to which populations, such as children and elders, are affected by climate-related disasters or viral outbreaks. Sensitivity is shaped by individual factors like age, disabilities, or pre-existing health conditions, urban factors such as access to basic services, housing conditions, and infrastructure. Adaptive capacity, or the ability to respond and recover, is influenced by levels of education, the presence of community organization, and the availability and quality of health and municipal services.

suicidal risks considering demographic factors, such as age and ethnicity, and support the use of a shorter version (PHQ-2) as an initial screening tool [77].

## Statistical analysis

This study applied a multi-step statistical approach to explore the associations between PSS and depression disorders with demographic, clinical, social factors, and neighborhood-related factors, guided by a vulnerability framework [6,18]. Prior to analysis, data validation procedures were conducted, including checks for completeness and internal consistency, with duplicate records removed using unique participant identifiers.

The statistical analysis consisted of three stages: Descriptive Statistics in which key characteristics of the sample were summarized according to the type of categorical and quantitative variables. Quantitative variables: Reported as mean, standard deviation (SD), median, and interquartile range (IQR). Categorical variables: expressed as frequencies and percentages. The distribution of stress and depression scores was examined across demographic, clinical, and socio-environmental variables to provide an overview of the sample characteristics.

## Correlation analysis

To assess the associations between stress, depression, and vulnerability dimensions, as well as to evaluate potential multicollinearity among vulnerability-related covariates, different correlation coefficients were applied depending on the nature of the variables: Kendall's tau-b (Tb): used for ordinal variables with similar scales to measure monotonic relationships, Kendall's tau-c (Tc): applied to ordinal variables with different category distributions and Cramér's V (Vc): selected for nominal categorical variables.

## Regression analysis

Demographic variables such as sex, age, ethnicity, housing conditions, access to drinking water and paved roads were considered part of the population's sensitivity in the urban environment. Variables such as education, economic activity, access to city services such as green areas, health and social services were considered adaptive capacity (or lack of) to face stress and depression disorders. A clinical condition related to COVID-19 testing, vaccination status, and preconditions was added in the analysis. Stress and depression were explored using a Spearman correlation and odds ratio analysis to assess the association between the independent and dependent variable with a $p < 0.05$ significance level. Multinomial logistic regression was used to calculate the adjusted odds ratio using formula (1).

$$\log\log \left( \frac{P(Y = k)}{P(Y = ref)} \right) = \beta_{0k} + \beta_{1k}X_1 + \beta_{2k}X_2 + \cdots + \beta_{nk}X_n, \qquad for\ k = 1,\ 2,\ \cdots, K-1$$

(1)

Where:

$P(Y = k)$ is the probability of the outcome being in category $k$. $P(Y = ref)$ is the probability of being in the reference category.

$\beta_{0k}$ is the intercept for outcome category $k$.

$\beta_{nk}$ are the coefficients for the independent variables $X_n$ in category $k$.

All statistical analyses were performed using R version 4.4.2. Data was validated prior to analyses, assumptions to each statistical test, including the scale and distribution of variables, were assessed. When assumptions of normality or homoscedasticity were not met, appropriate non-parametric techniques were applied. Missing data were not imputed; analyses were restricted to complete cases..

## Ethics approval

Research protocol was approved by the Expedited Ethics Committee of the National Directorate of Health Intelligence (DIS) of the Ministry of Public Health (MSP) (Ref No. 107–2021). Study Number 107–2020 approved on August 12, 2021. Participants were fully informed about the nature and purpose of the study in compliance with ethical standards for research, guaranteeing that participants were aware of their rights and the voluntary nature of their involvement.

## Results

### Socio-demographics variables

A total of 340 participants completed the two survey sections for PSS and depression. A total of 233 (68.5%) were women with a mean age of 44 years old (SD: 16.4) and men had a mean age of 47.8 years (SD: 18.3). Considering age range, the largest group was adults 229 (67%), followed by 71 young adults (18–29 years old) and teenagers (12–17 years old) (20.88%) and 40 seniors (> 65 years old) (11.76%). Regarding ethnicity, 76.4% of individuals self-identify as mestizo, 6.5% as "montubios" or countryside mestizos, 5.9% as white, 4.1% individuals as indigenous, and 14 (4.1%) as Afro-Ecuadorian 14 (4.1%). Of the total respondents, 166 (48.8%) individuals had. High school education, 102 (30%) s elementary education, 57 (16,7%) undergrad education, a total of 5 (1.5%) reported graduate studies, and 10 (2.9%) reported no education. Considering economic activity, 121 (31.6%) individuals were employed as workers (laborer), 114 (33.5%) were homemakers, 64 (18.8%) were unemployed, 29 (8.5%) were students, and 12 (3.5%) were retired. Regarding exposure-related variables, 77,4% of individuals had not traveled in the last month. In terms of COVID-19 diagnosis, 45,9% had not been tested, 13.2% had a positive result, and 39.7% were tested negative. The results of COVID-19 diagnostic test presumed that 121 (35,6%) individuals had the SARS-CoV2, and 219 (64,4%) were negative. Considering the COVID-19 vaccination, 82.4% were immunized.

The self-reported comorbidities among the individuals indicates that 3.5% had cancer, 18.2% a medical history of diabetes, and 9,4% had cardiovascular diseases, with the majority of these cases occurring in women, additionally 2.1% reported a medical history of COPD mostly men. Most of the individuals reported no drug consumption, and regarding alcohol consumption frequency 68.8% reported that they never had alcohol and 23.5% of the individuals just once a month or less.

Regarding the urban and built environment, 34.1% of individuals reported exposure to stagnant water. Water supply came from municipal pipelines for 50.8%, while 45.6% relied on water tanks—often the only option in many areas of Durán due to limited municipal capacity. Waste collection by garbage trucks was reported by 99% of respondents. Additionally, 31.8% lived on unpaved roads, which are typically associated with informal settlements and limited public services. See Table 1 for the sociodemographic and health characteristics stratified by sex.

## Self-reported stress

Self-reported level of moderate stress was mostly in women (66.2%), Individual vulnerability factors (sensitivity) were associated with sex, age, and relationship status. The variable sex is associated with self-reported stress (0.18, $p < 0.003$). Adults were significantly more likely to experience moderate levels of stress compared to adolescents (OR=9.49, 95% CI [1.47, 61.97], $p < 0.02$), indicating that adults have nearly 9.5 times higher odds of experiencing moderate stress levels. Participants in domestic partnerships demonstrated a significant protective effect against moderate stress levels (OR=0.2, 95% CI [0.06, 0.61], compared to singles, $p < 0.01$). This suggests that individuals who are in domestic partnership have 80% lower odds of experiencing moderate stress levels compared to those in domestic partnerships.

Urban vulnerability factors were associated to housing type, those participants living in collective housing arrangements demonstrated significantly higher stress levels compared to those living in individual houses: rented rooms in tenement housing: OR=41.26, 95% CI [2.02, >999], $p < 0.02$, and apartments: OR=46.53, 95% CI [2.74, >999], $p < 0.01$. This study also found a significant positive correlation between stress levels and depression symptoms (Kendall's tau = 0.42, p = 0.02), indicating that higher levels of stress were associated with more severe depression symptoms. Table 2. Describe the association of vulnerability factors with level of stress and depression and Table 3 presents the regression model predicting moderate stress; estimates for high stress were not reported due to very low event counts (<10) and limited interpretability.

## Depression

Individual vulnerability factors associated with depression were sex, pre-existing health conditions, COVID-19 immunization, and relationship status. Being a woman was significantly associated with mild depression: OR=4.44, 95% CI [1.54, 12.91], $p < 0.01$ moderately severe depression: OR=50.91, 95% CI [4.62, >999], p < 0.01. Sensitivity of pre-existing health conditions were associated with chronic obstructive pulmonary disease (COPD) ($V_c = 0.18$, $p < 0.003$). Considering the COVID-19 Status: not being vaccinated against COVID-19 was associated with higher depression levels Vc = 0.16, $p <= 0.043$, while negative COVID-19 test results were associated with moderate depression levels (OR=5.53, 95% CI [1.06, 28.55]). Considering relationship status; domestic partnership was associated with mild levels of depression (OR=0.21, 95% CI [0.06, 0.77], $p < 0.02$).

Urban vulnerability factors were associated to housing type, thus living in room(s) in tenement housing was associated with higher levels of depression (Tc = −0.06, $p <= 0.04$), and those living in tenement housing specifically showed strong association with mild depression (OR=77.48, 95% CI [2.53, >999], $p < 0.01$). Living in neighborhoods with unpaved roads was associated with higher depression levels overall and mild depression specifically (OR=2.89, 95% CI [1.0, 8.36], $p < 0.05$). Lack of access to water supply was associated with higher depression levels (Tc = −0.08, $p <= 0.038$). Table 4. shows the results of the regression model predicting levels of depression.

**Table 1. Sociodemographic and health characteristics stratified by sex.**

| Characteristic | Overall | | |
|---|---|---|---|
| | Total (N = 340) | Female (N = 233) | Male (N = 107) |
| **Age** | | | |
| Mean | 45.2 | 44 | 47.8 |
| Median (IQR) | 47 (32-57.3) | 44 (31-55) | 52 (35.5-62) |
| **Age Group** | | | |
| Adolescent | 22 (6.5%) | 10 (4.3%) | 12 (11.2%) |
| Youth | 49 (14.4%) | 40 (17.2%) | 9 (8.4%) |
| Adult | 229 (67.4%) | 159 (68.2%) | 70 (65.4%) |
| Senior | 40 (11.7%) | 24 (10.3%) | 16 (15.0%) |
| **Ethnicity** | | | |
| Afro-Ecuadorian | 14 (4.1%) | 7 (3.0%) | 7 (6.5%) |
| Mestizo | 259 (76.2%) | 183 (78.5%) | 76 (71.0%) |
| Indigenous | 14 (4.1%) | 12 (5.2%) | 2 (1.9%) |
| Montubio | 22 (6.5%) | 9 (3.9%) | 13 (12.1%) |
| Mulatto | 10 (2.9%) | 6 (2.6%) | 4 (3.7%) |
| White | 20 (5.9%) | 15 (6.5%) | 5 (4.7%) |
| No answer | 1 (0.3%) | 1 (0.4%) | 0 (0.0%) |
| **Educational Attainment** | | | |
| None | 10 (2.9%) | 6 (2.6%) | 4 (3.7%) |
| Elementary School | 102 (30.0%) | 72 (31.0%) | 30 (28.0%) |
| High School | 166 (48.8%) | 113 (48.5%) | 53 (49.5%) |
| Undergraduate | 57 (16.8%) | 39 (16.7%) | 18 (16.8%) |
| Graduate | 5 (1.5%) | 3 (1.3%) | 2 (1.9%) |
| **Marital Status** | | | |
| Single | 114 (33.6%) | 76 (32.6%) | 38 (35.5%) |
| Married | 107 (31.8%) | 67 (28.8%) | 40 (37.4%) |
| Domestic Partnership | 65 (19.1%) | 45 (19.3%) | 20 (18.7%) |
| Separated | 17 (5.0%) | 16 (6.9%) | 1 (0.9%) |
| Divorced | 15 (4.4%) | 10 (4.3%) | 5 (4.7%) |
| Widowed | 20 (5.9%) | 17 (7.3%) | 3 (2.8%) |
| No answer | 2 (0.6%) | 2 (0.8%) | 0 (0.0%) |
| **Occupational Status** | | | |
| Homemaker | 114 (33.6%) | 114 (49.0%) | 0 (0.0%) |
| Student | 29 (8.5%) | 18 (7.7%) | 11 (10.3%) |
| Worker | 121 (35.6%) | 66 (28.3%) | 55 (51.4%) |
| Retired | 12 (3.5%) | 4 (1.7%) | 8 (7.5%) |
| Unemployed | 64 (18.8%) | 31 (13.3%) | 33 (30.8%) |
| **Travel History (last 4 weeks)** | | | |
| Yes | 77 (22.6%) | 49 (21.0%) | 28 (26.2%) |
| No | 263 (77.4%) | 184 (79.0%) | 79 (73.8%) |
| **COVID-19 Diagnostic Test Outcome** | | | |
| Positive | 45 (13.2%) | 26 (11.1%) | 19 (17.8%) |
| Negative | 135 (39.7%) | 94 (40.3%) | 41 (38.3%) |
| No completed | 4 (1.2%) | 3 (1.3%) | 1 (0.9%) |
| No Tested | 156 (45.9%) | 110 (47.2%) | 46 (42.9%) |

*(Continued)*

**Table 1.** (Continued)

| Characteristic | Overall | | |
|---|---|---|---|
| | Total (N = 340) | Female (N = 233) | Male (N = 107) |
| **Presumed COVID-19** | | | |
| Yes | 121 (35.6%) | 88 (37.8%) | 33 (30.8%) |
| No | 219 (64.4%) | 145 (62.2%) | 74 (69.2%) |
| **COVID-19 Immunization** | | | |
| Yes | 280 (82.4%) | 189 (81.1%) | 91 (85.0%) |
| No | 60 (17.6%) | 44 (18.9%) | 16 (15.0%) |
| **Cancer Medical History** | | | |
| Yes | 12 (3.5%) | 11 (4.7%) | 1 (0.9%) |
| No | 321 (94.4%) | 218 (93.6%) | 103 (96.2%) |
| Do not know | 7 (2.0%) | 4 (1.7%) | 3 (2.8%) |
| **Diabetes Medical History** | | | |
| Yes | 62 (18.2%) | 40 (17.2%) | 22 (20.6%) |
| No | 267 (78.5%) | 185 (79.4%) | 82 (76.6%) |
| Do not know | 11 (3.2%) | 8 (3.4%) | 3 (2.8%) |
| **Cardiovascular Disease** | | | |
| Yes | 32 (9.4%) | 21 (9.0%) | 11 (910.3%) |
| No | 302 (88.8%) | 209 (89.7%) | 93 (86.9%) |
| Do not know | 6 (1.8%) | 3 (1.3%) | 3 (2.8%) |
| **COPD Medical History** | | | |
| Yes | 7 (2.1%) | 2 (0.9%) | 5 (4.7%) |
| No | 324 (95.3%) | 224 (96.1%) | 100 (93.4%) |
| Do not know | 9 (2.6%) | 7 (3.0%) | 2 (1.9%) |
| **Long-term Medical History** | | | |
| yes | 109 (32.1%) | 74 (31.8%) | 35 (32.7%) |
| No | 231 (67.9%) | 159 (68.2%) | 72 (67.3%) |
| **Drugs Consumption** | | | |
| Cannabis | 4 (1.2%) | 0 (0%) | 4 (3.8%) |
| Cocaine | 4 (1.2%) | 1 (0.4%) | 3 (2.8%) |
| Heroin | 2 (0.6%) | 0 (0.0%) | 2 (1.9%) |
| None | 327 (96.7%) | 231 (99.6%) | 96 (91.4%) |
| Other | 1 (0.3%) | 0 (0.0%) | 1 (0.9%) |
| No answer | 2 (0.6%) | 1 (0.04%) | 1 (0.9%) |
| **Alcohol Consumption Frequency** | | | |
| 1-2 times/week | 12 (3.6%) | 6 (2.6%) | 6 (5.7%) |
| 3-4 times/week | 2 (0.6%) | 1 (0.4%) | 1 (0.9%) |
| 2-3 times/month | 9 (2.7%) | 4 (1.7%) | 5 (4.7%) |
| Once month or less | 79 (23.5%) | 43 (18.5%) | 36 (33.6%) |
| Never | 231 (68.8%) | 172 (74.8%) | 59 (56.1%) |
| Do not know | 2 (0.6%) | 2 (0.8%) | 0 (0.0%) |
| No answer | 5 (1.5%) | 5 (2.2%) | 0 (0.0%) |
| **Paved Road** | | | |
| Yes | 218 (64.1%) | 148 (63.5%) | 70 (65.4%) |
| No | 108 (31.8%) | 76 (32.6%) | 32 (29.9%) |
| Do not know | 2 (0.6%) | 2 (0.9%) | 0 (0.0%) |

*(Continued)*

| Characteristic | Overall | | |
| --- | --- | --- | --- |
| | Total (N = 340) | Female (N = 233) | Male (N = 107) |
| No answer | 12 (3.5%) | 7 (3.0%) | 5 (4.7%) |
| **Housing Type** | | | |
| Hut/Shack | 12 (3.5%) | 7 (3.0%) | 5 (4.7%) |
| Emergency housing | 4 (1.2%) | 3 (1.3%) | 1 (0.9%) |
| Collective housing | 2 (0.6%) | 1 (0.4%) | 1 (0.9%) |
| Room(s) in tenement housing | 17 (5.0%) | 8 (3.4%) | 9 (8.4%) |
| Apartment | 25 (7.4%) | 18 (7.8%) | 7 (6.5%) |
| Private residence | 268 (78.8%) | 189 (81.1%) | 79 (73.8%) |
| Other private housing | 4 (1.2%) | 3 (1.3%) | 1 (0.9%) |
| No answer | 8 (2.3%) | 4 (1.7%) | 4 (3.7%) |
| **Stagnant Water** | | | |
| Yes | 116 (34.1%) | 78 (33.5%) | 38 (35.5%) |
| No | 210 (61.8%) | 145 (62.2%) | 65 (60.7%) |
| Do not know | 2 (0.6%) | 2 (0.9%) | 0 (0.0%) |
| No answer | 12 (3.5%) | 8 (3.4%) | 4 (3.7%) |
| **Waste Collection Method** | | | |
| Garbage is burned | 2 (0.6%) | 1 (0.4%) | 1 (0.9%) |
| Garbage truck | 328 (99.0%) | 226 (97.0%) | 102 (95.3%) |
| Other | 1 (0.3%) | 1 (0.4%) | 0 (0.0%) |
| No answer | 9 (2.6%) | 5 (2.1%) | 4 (3.7%) |
| **Water Supply Method** | | | |
| River/Spring/Canal | 1 (0.3%) | 0 (0.0%) | 1 (1.0%) |
| Well Water | 4 (1.2%) | 3 (1.3%) | 1 (1.0%) |
| Water Tanker | 151 (45.6%) | 108 (47.4%) | 43 (41.7%) |
| Municipal System | 168 (50.8%) | 114 (50.0%) | 54 (52.4%) |
| Other | 4 (1.2%) | 2 (0.9%) | 2 (1.9%) |
| Do not know | 3 (0.9%) | 1 (0.4%) | 2 (1.9%) |
| No answer | 9 (2.6%) | 5 (2.1%) | 4 (3.7%) |
| **Stress Level** | | | |
| Low | 94 (27.7%) | 55 (23.6%) | 39 (36.4%) |
| Moderate | 225 (66.2%) | 158 (67.8%) | 67 (62.6%) |
| High | 21 (6.2%) | 20 (8.6%) | 1 (0.9%) |
| **Depression Level** | | | |
| None/Minimal | 125 (36.8%) | 76 (32.6%) | 49 (45.8%) |
| Mild | 99 (29.1%) | 69 (29.6%) | 30 (28.0%) |
| Moderate | 65 (19.1%) | 45 (19.3%) | 20 (18.6%) |
| Moderately Severe | 35 (10.3%) | 30 (12.8%) | 5 (4.7%) |
| Severe | 16 (4.7%) | 13 (5.6%) | 3 (2.8%) |

[1]All values are n (%) unless otherwise specified. IQR = Interquartile Range.

Note: Percentages may not sum to 100% due to rounding.

COPD = Chronic Obstructive Pulmonary Disease.

**Table 2. Associations between sociodemographic, health, and environmental factors with stress and depression levels during the COVID-19 pandemic.**

| Vulnerability Dimension Variable | Stress Level | | Depression Level | |
|---|---|---|---|---|
| | Coefficient | *p-value* | Coefficient | *p-value* |
| **Sensitivity (Individual)** | | | | |
| Sex | 0.18 | ***0.003*** | 0.17 | ***0.049*** |
| Age Group | −0.03 | *0.492* | −0.01 | *0.709* |
| Ethnicity | −0.07 | *0.058* | −0.05 | *0.105* |
| Marital Status | 0.13 | *0.292* | 0.14 | *0.067* |
| Cancer Medical History | 0.06 | *0.602* | 0.14 | *0.109* |
| Long-Term Medical History | 0.05 | *0.637* | 0.19 | ***0.020*** |
| Diabetes Medical History | 0.06 | *0.812* | 0.12 | *0.207* |
| Cardiovascular Disease | 0.05 | *0.837* | 0.15 | ***0.018*** |
| COPD Medical History | 0.06 | *0.574* | 0.18 | ***0.003*** |
| Drugs Consumption | 0.08 | *0.795* | 0.11 | *0.374* |
| Alcohol Consumption | −0.02 | *0.587* | −0.06 | *0.107* |
| **Sensitivity (Urban)** | | | | |
| Paved Road | 0.06 | *0.214* | 0.16 | ***0.005*** |
| Housing Structure | −0.06 | *0.082* | −0.06 | *0.088* |
| Stagnant Water | 0.09 | *0.304* | 0.10 | *0.388* |
| Waste Collection | 0.04 | *0.782* | 0.10 | *0.515* |
| Water Supply | −0.01 | *0.594* | −0.08 | ***0.036*** |
| **Adaptive Capacity** | | | | |
| Education Level | −0.03 | *0.570* | 0.09 | *0.101* |
| Occupational Status | 0.13 | *0.920* | 0.09 | *0.933* |
| COVID-19 Immunization | 0.10 | *0.156* | 0.16 | ***0.043*** |
| **Exposure** | | | | |
| Travel History (last 4 weeks) | 0.09 | *0.232* | 0.10 | *0.561* |
| COVID-19 Test Result | 0.10 | *0.275* | 0.12 | *0.266* |
| Presumed COVID-19 | 0.03 | *0.815* | 0.05 | *0.946* |

**Note:** Coefficients and *p-values* from association analyses. Statistical tests: $T_b$ = Kendall's tau-b, $T_c$ = Kendall's tau-c, $Vc$ = Cramér's V. Significant associations ($p < 0.05$) are highlighted in bold.

COPD = Chronic Obstructive Pulmonary Disease.

## Discussion

Our findings align with global research indicating that mental health vulnerabilities in the Global South are shaped by a confluence of individual, environmental, economic, and systemic factors. This understanding aims to guide interventions and support policy integration of disaster risk reduction and health emergency management in cities within the Global South [78,79]. Individual sensitivity factors such as gender, age, and chronic illness; urban infrastructure deficiencies, like informal settings, unpaved roads and inadequate housing, and limited adaptive capacities such as vaccination access and health care are all risk factors that can raise mental health issues during disasters [21,80]. These findings call for multi-level interventions that recognize the interplay between physical infrastructure, public health, and social systems in shaping mental well-being [81].

### Socio-cultural Vulnerability

Gender functions as a key vulnerability factor in health emergencies, with women in our study showing significantly higher likelihood of experiencing both stress and depression compared to men. This aligns with global patterns observed before

**Table 3. Multivariable logistic regression of stress level predictors.**

| Vulnerability | | Moderate Stress | |
|---|---|---|---|
| **Dimension** | **Variable** | **OR [95% CI]** | *P-value* |
| **Sensitivity (Individual)** | Sex | | |
| | Male (ref) | 1 | – |
| | Female | 1.22 [0.52 - 2.82] | *0.65* |
| | Age Group | | |
| | Adolescent (ref) | 1 | – |
| | Youth | 1.95 [0.36 - 10.51] | *0.44* |
| | Adult | 9.49 [1.47 - 61.97] | ***0.02*** |
| | Senior | 4.71 [0.47 - 47.51] | *0.19* |
| | Ethnicity | | |
| | Afro/Ecuadorian (ref) | 1 | – |
| | Mestizo | 0.70 [0.11 - 4.71] | *0.72* |
| | Indigenous | 0.92 [0.07 - 11.96] | *0.95* |
| | Montubio | 0.11 [0.01 - 1.04] | *0.06* |
| | Mulatto | 0.36 [0.03 - 4.54] | *0.43* |
| | White | 0.25 [0.02 - 2.63] | *0.25* |
| | Marital Status | | |
| | Single (ref) | 1 | – |
| | Married | 0.36 [0.12 - 1.13] | *0.08* |
| | Domestic Partnership | 0.20 [0.06 - 0.61] | ***0.01*** |
| | Separated | 0.79 [0.11 - 5.46] | *0.81* |
| | Divorced | 0.29 [0.05 - 1.64] | *0.16* |
| | Widowed | 0.41 [0.07 - 2.46] | *0.33* |
| | Cancer Medical History | | |
| | Yes (ref) | 1 | – |
| | No | 2.23 [0.76 - 6.51] | *0.15* |
| | Long-Term Medical History | | |
| | Yes (ref) | 1 | – |
| | No | 0.84 [0.12 - 5.87] | *0.86* |
| | Diabetes Medical History | | |
| | Yes (ref) | 1 | – |
| | No | 0.52 [0.20 - 1.34] | *0.18* |
| | Cardiovascular Disease Medical History | | |
| | Yes (ref) | 1 | – |
| | No | 0.97 [0.29 - 3.32] | *0.97* |
| | COPD Medical History | | |
| | Yes (ref) | 1 | – |
| | No | 1.26 [0.18 - 9.07] | *0.82* |
| | Drugs Consumption | | |
| | Cannabis (ref) | 1 | – |
| | Cocaine | 0.73 [0.02 - 30.42] | *0.87* |
| | None | 2.12 [0.12 - 37.83] | *0.61* |
| | Alcohol Consumption Frequency | | |
| | Never (ref) | 1 | – |
| | 1-2 times/week | 0.94 [0.16 - 5.45] | *0.95* |
| | Once month or less | 0.98 [0.43 - 2.25] | *0.97* |
| | 2-3 times/month | 0.64 [0.08 - 5.17] | *0.67* |

*(Continued)*

**Table 3.** (Continued)

| Vulnerability | | Moderate Stress | |
| --- | --- | --- | --- |
| Dimension | Variable | OR [95% CI] | *P-value* |
| **Sensitivity (Urban)** | Housing Type | | |
| | Hut/Shack (ref) | 1 | – |
| | Emergency housing | 0.38 [0.39 - 39.46] | *0.68* |
| | Private Residence | 5.87 [0.71 - 48.75] | *0.10* |
| | Paved Road | | |
| | Yes (ref) | 1 | – |
| | No | 1.11 [0.42 - 2.87] | *0.84* |
| | Stagnant Water | | |
| | Yes (ref) | 1 | – |
| | No | 0.61 [0.25 - 1.49] | *0.28* |
| **Adaptive Capacity** | Educational Attainment | | |
| | None (ref) | 1 | – |
| | Elementary School | 0.68 [0.07 - 7.06] | *0.75* |
| | High School | 2.25 [0.21 - 23.84] | *0.50* |
| | Undergraduate | 1.25 [0.11 - 14.17] | *0.86* |
| | Graduate | 0.47 [0.01 - 17.92] | *0.68* |
| | Occupational Status | | |
| | Homemaker (ref) | 1 | – |
| | Student | 1.31 [0.22 - 7.99] | *0.77* |
| | Worker | 0.59 [0.22 - 1.59] | *0.30* |
| | Retired | 0.47 [0.06 - 3.50] | *0.46* |
| | Unemployed | 0.86 [0.27 - 2.75] | *0.80* |
| | COVID-19 Immunization | | |
| | Yes (ref) | 1 | – |
| | No | 2.23 [0.76 - 6.51] | *0.15* |
| **Exposure** | Travel History (last 4 weeks) | | |
| | Yes (ref) | 1 | – |
| | No | 1.49 [0.68 - 3.31] | *0.32* |
| | Covid-19 Diagnostic Test Outcome | | |
| | Positive (ref) | 1 | – |
| | Negative | 2.12 [0.72 - 6.15] | *0.17* |
| | No Completed | 0.62 [0.05 - 8.02] | *0.71* |
| | No Tested | 1.58 [0.56 - 4.48] | *0.39* |
| | Presumed COVID-19 | | |
| | Yes (ref) | 1 | – |
| | No | 0.64 [0.30 - 1.36] | *0.25* |

Note: CI = Confidence Interval, OR = Odds Ratio adjusted, ref = reference group. All p-values are shown in italics, with statistically significant values (p < 0.05) in bold. COPD = Chronic Obstructive Pulmonary Disease.

the pandemic, where depression rates were consistently higher among women compared to men (5.8% versus 3.8%, respectively), indicating pre-existing gender disparities in psychosocial health that crises and disasters tend to exacerbate rather than create [4,82].

In Ecuador, women face multiple intersecting vulnerabilities that explain these disparities. Before the pandemic, 8.1% of women aged 15–49 years reported experiencing physical or sexual violence from current or former intimate partners,

**Table 4. Multivariable logistic regression of mild and moderate depression level predictors by vulnerability dimension.**

| Vulnerability | | Mild | | Moderate | |
|---|---|---|---|---|---|
| Dimension | Variable | OR (95% CI) | *P-value* | OR (95% CI) | *P-value* |
| **Sensitivity (Individual)** | Sex | | | | |
| | Male | 1.00 (ref) | – | 1.00 (ref) | – |
| | Female | 4.44 (1.54-12.91) | **0.01** | 1.62 (0.49-5.30) | 0.43 |
| | Age Group | | | | |
| | Adolescent | 1.00 (ref) | – | 1.00 (ref) | – |
| | Youth | 0.64 (0.09-4.83) | 0.67 | 3.71 (0.28-49.57) | 0.32 |
| | Adult | 0.38 (0.04-3.26) | 0.38 | 5.58 (0.36-86.03) | 0.22 |
| | Senior | 0.16 (0.01-2.20) | 0.17 | 0.63 (0.02-17.88) | 0.79 |
| | Ethnicity | | | | |
| | Afro/Ecuadorian | 1.00 (ref) | – | 1.00 (ref) | – |
| | Mestizo | 0.04 (0.01-0.53) | **0.01** | – | **–** |
| | Indigenous | 0.04 (0.01-0.82) | **0.04** | – | **–** |
| | Montubio | 0.01 (0.00-0.19) | **0.00** | – | **–** |
| | Mulatto | 0.02 (0.00-0.48) | **0.02** | – | **–** |
| | White | 0.03 (0.00-0.52) | **0.02** | – | **–** |
| | Marital Status | | | | |
| | Single | 1.00 (ref) | – | 1.00 (ref) | – |
| | Married | 0.33 (0.10–1.06) | 0.06 | 0.34 (0.08–1.38) | 0.13 |
| | Domestic Partnership | 0.21 (0.06-0.77) | **0.02** | 0.45 (0.11-1.82) | 0.26 |
| | Separated | 0.24 (0.02-2.51) | 0.23 | 0.72 (0.09–5.71) | 0.76 |
| | Divorced | 0.61 (0.07-5.42) | 0.66 | 1.09 (0.09-13.28) | 0.94 |
| | Widowed | 0.49 (0.07–3.52) | 0.48 | 0.95 (0.09–9.92) | 0.97 |
| | Cancer Medical History | | | | |
| | Yes | 1.00 (ref) | – | 1.00 (ref) | – |
| | No | 0.53 (0.04-7.13) | 0.63 | 0.53 (0.04-6.67) | 0.62 |
| | Long-Term Medical History | | | | |
| | Yes | 1.00 (ref) | – | 1.00 (ref) | – |
| | No | 0.22 (0.08-0.66) | **0.01** | 0.45 (0.14-1.45) | 0.18 |
| | Diabetes Medical History | | | | |
| | Yes | 1.00 (ref) | – | 1.00 (ref) | – |
| | No | 0.83 (0.26-2.66) | 0.75 | 0.43 (0.13-1.48) | 0.18 |
| | Cardiovascular Disease | | | | |
| | Yes | 1.00 (ref) | – | 1.00 (ref) | – |
| | No | 2.53 (0.50-12.70) | 0.26 | 0.79 (0.17-3.61) | 0.77 |
| | Drugs Consumption | | | | |
| | Cannabis | 1.00 (ref) | – | 1.00 (ref) | – |
| | None | 0.19 (0.00- 7.97) | 0.38 | 0.11 (0.00-3.11) | 0.19 |
| | Alcohol Consumption | | | | |
| | Never | 1.00 (ref) | – | 1.00 (ref) | – |
| | 1-2 times/week | 0.00 | – | 0.47 (0.05-4.13) | 0.50 |
| | Once month or less | 0.58 (0.22-1.52) | 0.27 | 0.43 (0.14-1.34) | 0.15 |
| | 2-3 times/month | 0.39 (0.04-4.27) | 0.44 | 0.00 | – |

*(Continued)*

**Table 4.** (Continued)

| Vulnerability | | Mild | | Moderate | |
|---|---|---|---|---|---|
| Dimension | Variable | OR (95% CI) | *P-value* | OR (95% CI) | *P-value* |
| **Sensitivity (Urban)** | Housing Type | | | | |
| | Hut/Shack | 1.00 (ref) | – | 1.00 (ref) | – |
| | Private Residence | 3.19 (0.20-51.22) | *0.41* | 0.23 (0.03-1.88) | *0.47* |
| | Paved Road | | | | |
| | Yes | 1.00 (ref) | – | 1.00 (ref) | – |
| | No | 2.89 (1.00-8.36) | *0.05* | 4.44 (1.33-14.71) | *0.42* |
| | Stagnant Water | | | | |
| | Yes | 1.00 (ref) | – | 1.00 (ref) | – |
| | No | 0.74 (0.29-1.87) | 0.52 | 1.60 (0.53-4.80) | 0.18 |
| **Adaptive Capacity** | Educational Attainment | | | | |
| | None | 1.00 (ref) | – | 1.00 (ref) | – |
| | Elementary School | 0.20 (0.01-4.95) | *0.32* | 0.02 (0.00-0.47) | ***0.02*** |
| | High School | 0.19 (0.01-4.67) | *0.31* | 0.03 (0.00-0.79) | ***0.03*** |
| | Undergraduate | 0.12 (0.00-3.18) | *0.20* | 0.01 (0.00-0.34) | ***0.01*** |
| | Occupational Status | | | | |
| | Homemaker | 1.00 (ref) | – | 1.00 (ref) | – |
| | Student | 0.15 (0.02-1.26) | *0.08* | 1.99 (0.18-22.02) | *0.57* |
| | Worker | 1.04 (0.36-2.99) | *0.94* | 1.28 (0.36-4.57) | *0.70* |
| | Unemployed | 1.12 (0.29-4.33) | *0.88* | 1.55 (0.32-7.45) | *0.58* |
| | COVID-19 Immunization | | | | |
| | Yes | 1.00 (ref) | – | 1.00 (ref) | – |
| | No | 0.95 (0.29-3.18) | *0.94* | 5.87 (1.51-22.76) | ***0.01*** |
| **Exposure** | Travel History | | | | |
| | Yes | 1.00 (ref) | – | 1.00 (ref) | – |
| | No | 1.34 (0.54-3.33) | *0.53* | 1.99 (0.67-5.98) | *0.21* |
| | COVID-19 Diagnostic Test Outcome | | | | |
| | Positive | 1.00 (ref) | – | 1.00 (ref) | – |
| | Negative | 0.62 (0.16-2.45) | *0.50* | 5.53 (1.06-28.55) | ***0.04*** |
| | No Tested | 1.58 (0.44-5.64) | *0.48* | 4.10 (0.86-19.49) | *0.08* |
| | Presumed COVID-19 | | | | |
| | Yes | 1.00 (ref) | – | 1.00 (ref) | – |
| | No | 1.84 (0.75-4.50) | *0.18* | 0.68 (0.25-1.86) | *0.46* |

**Note:** OR = Odds Ratio; CI = Confidence Interval; ref = reference category. All p-values are shown in italics, with statistically significant values (p < 0.05) in bold.

COPD = Chronic Obstructive Pulmonary Disease.

establishing gender-based violence as a common public health problem and systemic risk for women in Ecuador [83]. The COVID-19 lockdowns intensified this vulnerability, with Ecuador experiencing a documented increase in gender-based violence during confinement periods [84]. Beyond violence, gendered cultural expectations impose additional burdens that heighten vulnerability during crises. Ecuadorian women and girls over 15 years old spend 20.9% of their time on unpaid work, compared to just 4.8% for men [85]. These disparities reflect deep-rooted social roles where women are responsible for the health and wellbeing of children and the elderly, household management, and often engage in small income

entrepreneurship while balancing domestic responsibilities [86–88]. As schools transitioned to virtual learning during the COVID-19 pandemic, women experienced increased caregiving demands, balancing their children's educational needs, household responsibilities, and for many, their own remote jobs [89,90]. This "triple burden" of paid work, unpaid domestic labor, and educational support compounded stressors that manifested in the higher depression and stress levels observed in Durán and in the Latin America region [91]. The disproportionate psychosocial impacts experienced by women during the COVID-19 pandemic reveal how gender-based vulnerabilities operate through specific mechanisms including increased care burdens, exposure to violence, limited resource access, and cultural expectations [89].

Considering age groups, while adults aged 30–50 reported moderate stress levels compared to younger adults, no significant link was observed with depression. These findings differ from other studies, which often show stronger associations between younger age and mental health outcomes [92,93]. The stress reported by adults in our study may reflect a complex combination of emotional strain, physical vulnerability, and heightened responsibilities, particularly in managing work, family, and financial challenges during the pandemic [94]. Financial instability due to job loss or income reduction likely contributed to increased stress in this age group, especially in Ecuador, where social safety nets are limited.

Regarding relationships status, individuals in domestic partnerships reported lower levels of moderate stress compared to single individuals, although no significant association was observed with depression. Relationship quality has been shown to be beneficial for mental health, with those in good-quality relationships having better mental health than those who are single, and significant differences noted in studies conducted in Austria and Canada [21,95–97]. During the pandemic, groups who self-identified as having partners had lower mean PHQ-9 scores than those who were single [98]. Research in Ecuador has suggested social isolation caused by the pandemic negatively impacted mental health [99].

## Health related vulnerability

Our analysis revealed higher depression levels among individuals with chronic obstructive pulmonary disease (COPD), although no clear association with stress was identified. Social distancing measures intensified feelings of isolation and loneliness for this vulnerable group and potentially limited their access to critical support systems [100]. Decreased overall physical health, the fear of care denial during healthcare system overload, combined with stigma, created additional psychological burdens [11,101]. These findings highlight the need for specialized psychosocial health support for individuals with pre-existing conditions and education about COPD within pulmonary rehabilitation programs to prepare for future crises.

Unvaccinated individuals exhibited higher levels of depression, highlighting a complex relationship between health behaviors, risk perception, and mental well-being [102]. According to national vaccination statistics, 85.23% of residents in Guayas, where Durán is located, had received a full course of COVID-19 vaccination by the end of 2022 [62,103]. Previous studies report mixed evidence regarding the association between depression and COVID-19 vaccination, with some findings suggesting lower vaccine uptake among individuals with depressive symptoms [104,105]. On the other hand, in Durán, Orellana et al. (2026) observed high willingness to follow COVID-19 recommendations and vaccination guidance, reflecting strong trust in local authorities and health providers despite limited disease knowledge transmission [62]. Suggesting that depressive symptoms, rather than information deficits, may shape individual responses; however, the present study is designed to examine associations rather than to establish causal relationships.

Factors such as age, gender, education level, limited knowledge about the virus, and low trust in the health system may contribute to vaccine resistance. This creates a concerning feedback loop, where reduced engagement in protective behaviors may heighten mental health risks [83,84]. These findings are particularly relevant for public health communication strategies in Global South contexts, where vaccine hesitancy often intersects with limited healthcare access, low health literacy, and culturally rooted beliefs [35,106].

## Urban vulnerability

Research on vulnerability in Durán indicated that people of various ages and family members often share the same plot [59]. These high-density living conditions hindered compliance to social distancing guidelines, increasing stress about potential transmission from neighbors [107]. Nonetheless, studies in Latin America argue that community housing can foster latent communal bonds, many of which were activated during the pandemic and climate emergencies [108].

People living in unpaved roads areas were associated with mild depression levels highlighting how basic infrastructure deficiencies can directly impact psychosocial health during health and climate hazards. Poor accessibility limits mobility and access to essential services, increasing isolation and anxiety [109]. Informal settlements in the Latin America region, including inadequate infrastructure, unpaved roads, overcrowding, and poor sanitation can exacerbate stress and depression, contributing to a complex web of health risks and socioeconomic challenges [110,111]. Climate-related hazards and health crises share parallel dynamics that both restrict mobility, increase isolation, and strain resources, intensifying stress and depression in affected communities [53,54,58,59]. The dual burden of natural disasters and the pandemic, and deficiencies in physical infrastructure create cascading impacts on mental health during crises [12].

Duran's study contributes to intersectionality research by moving beyond the individual level to examine how structural and environmental inequities at the urban scale shape vulnerability [112,113]. Showing that gender, age, health status, inadequate urban infrastructure, residence in informal neighborhoods, and access to knowledge and health information interact to produce distinct patterns of vulnerability that help explain mental health disparities during health-related crises. Besides urban power asymmetries, violence, and the availability of safe spaces emerge as critical contextual factors for the design and implementation of effective interventions [114].

## A Pathway to building resilient communities

Community resilience has been defined as the capacity to prepare for anticipated hazards, adapt to changing conditions, and withstand and recover rapidly from disturbances related to climate, health, and other crises [115,116]. Recovery from COVID-19 impacts and climate hazards requires a holistic approach that extends beyond immediate disaster response phases, providing sustained mental and psychosocial health support tailored to community needs to build resilience. This necessitates understanding pre-disaster vulnerabilities, health information, resource access, and capacities at the city level [36].

Durán's Risk Management Office in partnership with Academia and other stakeholders has been working to organize the First Responders Community Brigades (CB), formed by volunteers from the local neighborhoods trained in first aid, rescue techniques, and fire response. Brigade members include community leaders, women, and individuals of varying ages, skills, and knowledge. These brigades can play a critical role in mapping social vulnerability by incorporating psychosocial dimensions, identifying groups at higher physical or mental risk, as well as empowering communities to engage in learning activities for preparedness to recovery in health crises and climate-related disasters. The strategies may include psychosocial first aid, skill-building workshops, family communication plans, and preparedness kits (including children's items), to foster safety, support, and resilience before disasters occur. Strengthening community social networks at community level, simultaneously builds resilience to both health emergencies and climate hazards [117]. The Entre Nosotras initiative illustrates culturally adapted, scalable strategies for Ecuador and other Latin American cities by shifting from individual to group-based interventions aligned with community preferences, incorporating shared childcare arrangements, scheduling activities during evenings and weekends, and strengthening safe community spaces to support psychosocial wellbeing [118].

Post-disaster recovery is a vital aspect of public health response and preparation for future mental health challenges resulting from disasters and health emergencies [38,119]. Successful post-disaster recovery requires enhancing the adaptive capacity of communities through integrated Health Emergency and Disaster Risk Management Frameworks that address long-term psychosocial needs [38]. Community-specific strategies are fundamental to effective post-disaster

recovery, a "build back better" approach that integrates mental health support into disaster preparedness [120]. For example, the Community-Based Disaster Mental Health Intervention (CBDMHI) in Australia successfully increased disaster preparedness, resilience, and social cohesion while reducing symptoms of depression and anxiety [121]. Similarly, culturally adapted, group-based interventions facilitated by trained lay mental health workers in Haiti improved both disaster preparedness and mental health outcomes [122]. Community groups organized to work together can effectively address mental health challenges, leading to improved outcomes during crises [123,124]. Moreover, psychoeducation literacy programs can recognize early psychosocial health issues facilitating timely community support [125]. Programs can be tailored to empower women and girls with psychoeducation, promoting women's leadership and decision making (ex; at community brigades), gender-based violence services with safe places, and disaster risk management training [118,126,127].

### Integrating health care, mental health systems, and disaster risk management

A key step in this integration is establishing a clear national policy that promotes coordination, information sharing, and planning among healthcare, mental health, and disaster risk management institutions. Ecuador has made progress in this area. The Ecuadorian Humanitarian System recognizes the importance of "mental health and psychosocial support" (MHPSS), as reflected in instruments like the "Mental Health Strategic Operation Protocol in Emergencies due to COVID-19," developed in 2020 and updated in 2021 by the health and pre-care task group and the Ecuadorian Health Ministry [38]. However, implementation barriers remain that inhibit optimal benefits, including limited integration of public health policy with disaster risk reduction, insufficient human resources, institutional capacity, and financial constraints at the local level.

Despite these challenges, opportunities exist for advancing integration efforts. Studies emphasize the importance of strengthening community mental health services through collaborative care strategies, regular training for healthcare providers, robust quality assurance mechanisms, and accessible support networks [16,128]. Partnerships, as seen in Durán City, with involvement of different stakeholders such as academia, NGOs, and city-health practitioners, lead to key opportunities to integrate multi-level policies. During disaster response and recovery, monitoring self-care strategies among practitioners and addressing barriers faced by first responders is essential for fostering resilience [15,129]. These measures not only address immediate psychosocial health concerns but also promote long-term mental wellbeing that enhances community resilience [1,6].

### Strengths and limitations

One of our strengths is the knowledge of the vulnerability and hazards of Duran city, through previous studies since 2018, that acknowledge the underlying causes that exacerbate stress and depression during health crises and disasters. In addition, partnerships established with DRM officers, the Duran municipality, and community-based organizations, have been key enabling factors to develop this integrated study.

While this study offers valuable insights into the vulnerability factors associated with stress and depression in urban settings, several limitations must be acknowledged. The cross-sectional nature of the study limits the ability to establish causality between vulnerability factors and mental health outcomes. Data was collected through self-reported measures, which may be influenced by recall errors or social desirability bias, potentially affecting the accuracy of the responses. While the Cronbach's alpha coefficient for the stress scale was within acceptable limits, future studies should aim to strengthen the reliability and validity of all mental health indicators used.

Finally, although the sample size obtained (n = 340) provided adequate statistical power to detect associations of moderate magnitude, some associations, particularly those involving less frequent categories, were estimated with wide confidence intervals and non-significant results. This reflects limited precision due to data scarcity rather than the absence of an underlying association, which has been considered in the interpretation of findings.

## Conclusions and future work

The findings from Durán study highlight the critical need of an intersectional perspective to integrate psychosocial health, health care, and disaster risk management, particularly in urban settings marked by vulnerability and systemic inequalities. City, health care, social and emergency systems must be synergized to create better opportunities to help the most vulnerable populations and to enhance resilience during emergencies and disasters. The COVID-19 pandemic in Durán, Ecuador, revealed the multifaceted nature of psychosocial risk, shaped by individual sensitivity factors (such as gender, age, and chronic illness), urban infrastructure deficiencies (like informal settings, unpaved roads and inadequate housing), and limited adaptive capacities (vaccination access and health care). These findings call for multi-level interventions that recognize the interplay between physical infrastructure, public health, and social systems in shaping mental well-being.

Moving forward, disaster risk reduction must adopt an intersectoral, multi-hazard approach. Efforts to 'build back better' should prioritize reducing vulnerabilities in mental and psychosocial health, ensuring equitable access to health services, and promoting inclusive urban planning integrated into local disaster risk management policies.

Future research should focus on several key areas to strengthen the integration of psychosocial health within disaster risk management, such as:

- Exploring how to incorporate psychosocial support before, during, and after disasters within all four components of multi-hazard Early Warning Systems (EWS): vulnerability and risk analysis, monitoring and alert, dissemination and communication, and response. Community engagement would be key in this goal.

- Building up a database of psychological impacts along with the assessment of local mental health interventions, to better understand the processes that occur under compound risk and cascade events. This will allow us to improve current policies of mental health and disaster risk management in cities.

- Developing applied research that bridges the gap between academic findings and policy implementation, with an analysis of the stakeholders and the local regulatory frames to improve policy design. Partnerships between academic and research centers, city authorities, and national DRM institutions, and public health officers is a strategy that can enhance capacities and close this gap.

Only through such holistic integration can communities like Durán become more resilient, equitable, and prepared for the complex crises of the future.

## Data sharing

No individual participant data is available due to national policies on personal data protection. The steering committee will consider requests for clarification on specific issues related to the current publication, provided that sharing such data does not interfere with future publications by the investigators. This study complies with the requirements of Ecuador's Organic Law on Personal Data Protection.

## Supporting information

**S1 File. a) Durán survey questionnaire and instruments used to assess b) Perceived stress and c) Depression.**
(PDF)

**S2 File. Ethical approval issued by Ministry of Health of Ecuador (Spanish version and English translation).**
(PDF)

## Acknowledgments

We would like to acknowledge the interinstitutional coordination of Rommel Caiza, Cristian Torres, Virgilio Benavides, and Joyzette Mendoza, the technological assistance of Mauricio Leiton, the logistical fieldwork support of Eduardo Quimi, to Washington B. Cárdenas of the Laboratory for Biological Assays, Biologicals and Diagnostics-ENBIDIA, for the COVID-19 tests; and the administrative support of the Pacific International Center for Disaster Risk Reduction at ESPOL.

## Author contributions

**Conceptualization:** Mercy J. Borbor-Cordova, Katty Castillo, Christina D Campagna.

**Data curation:** Heydi Roa-Lopez.

**Formal analysis:** Mercy J. Borbor-Cordova, Heydi Roa-Lopez, Andrea Orellana-Manzano.

**Funding acquisition:** Mercy J. Borbor-Cordova.

**Investigation:** Mercy J. Borbor-Cordova, Madison Searles, Heydi Roa-Lopez, Andrea Orellana-Manzano, Christina D Campagna.

**Methodology:** Mercy J. Borbor-Cordova, Heydi Roa-Lopez, Katty Castillo, Andrea Orellana-Manzano, Christina D Campagna.

**Project administration:** Mercy J. Borbor-Cordova, Maria del Pilar Cornejo-Rodriguez.

**Resources:** Maria del Pilar Cornejo-Rodriguez.

**Supervision:** Mercy J. Borbor-Cordova, Andrea Orellana-Manzano, Christina D Campagna.

**Validation:** Maria del Pilar Cornejo-Rodriguez, Andrea Orellana-Manzano.

**Visualization:** Madison Searles, Andrea Orellana-Manzano.

**Writing – original draft:** Mercy J. Borbor-Cordova, Madison Searles, Christina D Campagna.

**Writing – review & editing:** Mercy J. Borbor-Cordova, Madison Searles, Heydi Roa-Lopez, Maria del Pilar Cornejo-Rodriguez, Katty Castillo, Andrea Orellana-Manzano, Christina D Campagna.

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
