## [Decision Letter · Decision Letter 0]

12 Feb 2025

PONE-D-24-34167Integrating Psychosocial Health into Disaster Risk Management: Insights from COVID-19 in Duran, Ecuador.PLOS ONE

Dear Dr. Borbor-Cordova,

Thank you for submitting your manuscript to PLOS ONE. After careful consideration, we feel that it has merit but does not fully meet PLOS ONE’s publication criteria as it currently stands. Therefore, we invite you to submit a revised version of the manuscript that addresses the points raised during the review process.

We look forward to receiving your revised manuscript.

Kind regards,

Ian Christopher N Rocha, MD, MBA, MHSS

Academic Editor

PLOS ONE

Journal Requirements:

“Research reported in this publication was supported by the Pacific International Center for Disaster Risk Reduction at ESPOL (CIPRRD-03-2021), Secretary Risk Management of Ecuador, and the support on the field campaign by the Municipality of Duran.”

5. We note that you have indicated that there are restrictions to data sharing for this study. PLOS only allows data to be available upon request if there are legal or ethical restrictions on sharing data publicly. For more information on unacceptable data access restrictions, please see http://journals.plos.org/plosone/s/data-availability#loc-unacceptable-data-access-restrictions.

6. We note that Figure 1 in your submission contain map/satellite images which may be copyrighted. All PLOS content is published under the Creative Commons Attribution License (CC BY 4.0), which means that the manuscript, images, and Supporting Information files will be freely available online, and any third party is permitted to access, download, copy, distribute, and use these materials in any way, even commercially, with proper attribution. For these reasons, we cannot publish previously copyrighted maps or satellite images created using proprietary data, such as Google software (Google Maps, Street View, and Earth). For more information, see our copyright guidelines: http://journals.plos.org/plosone/s/licenses-and-copyright.

Additional Editor Comments:

Please address the comments and suggestions of all reviewers. Thank you.

Reviewers' comments:

Reviewer's Responses to Questions

**Comments to the Author**

1. Is the manuscript technically sound, and do the data support the conclusions?

Reviewer #1: Yes

Reviewer #2: No

Reviewer #3: Yes

Reviewer #4: Yes

Reviewer #5: Yes

Reviewer #6: Partly

Reviewer #7: Partly

Reviewer #8: Yes

Reviewer #9: Yes

2. Has the statistical analysis been performed appropriately and rigorously? 

Reviewer #1: Yes

Reviewer #2: I Don't Know

Reviewer #3: Yes

Reviewer #4: I Don't Know

Reviewer #5: Yes

Reviewer #6: No

Reviewer #7: Yes

Reviewer #8: Yes

Reviewer #9: Yes

3. Have the authors made all data underlying the findings in their manuscript fully available?

Reviewer #1: Yes

Reviewer #2: Yes

Reviewer #3: Yes

Reviewer #4: No

Reviewer #5: Yes

Reviewer #6: Yes

Reviewer #7: No

Reviewer #8: Yes

Reviewer #9: Yes

4. Is the manuscript presented in an intelligible fashion and written in standard English?

Reviewer #1: Yes

Reviewer #2: Yes

Reviewer #3: Yes

Reviewer #4: Yes

Reviewer #5: Yes

Reviewer #6: Yes

Reviewer #7: No

Reviewer #8: No

Reviewer #9: Yes

5. Review Comments to the Author

Reviewer #1: 1. Subheading 3.1 - The statistical methods should be explained with procedures, interpretations, and equations/formulas.

2. Discuss these articles which focused on disaster vulnerability.

https://link.springer.com/article/10.1007/s10668-023-04149-1

https://link.springer.com/article/10.1007/s00477-022-02267-2

3. The subheading numbering should be properly arranged.

4. The results should be justified with the support of relevant published articles.

5. The influence of all variables (mentioned in the Tables) on disaster vulnerability should be explained in the Methods section.

6. Improve the quality of English to rectify the typo and grammatical errors.

7. Keywords - Include only 5 keywords which are relevant.

Reviewer #2: 1. Photos are not clear enough

2. Please give a brief description of the study area Duran such as population, economy, orientation, etc. to help the reader understand it

3. The introduction is too long and not very relevant. The methodological content is too simple and some necessary methodologies are not considered

4.221-223 grading criteria please add references

5. P-value should be italicized

6. How was the independent variable selected?

7. Is OR corrected and how are confounders considered?

8. How was sample size calculation considered?

Reviewer #3: Introduction

Please bring the following items

1- Definition of the research problem

2- The magnitude and importance of the study variable

3- Expressing the necessity of conducting the study

Finally, the practical purpose of the study should be stated

Methods

1.The method section should include the sample size formula.

2. Exclusion and inclusion criteria are fully stated.

3. The author did not mention any incomplete, incorrect or corrupted questionnaire.

4. The first paragraph of the discussion should be the overall result of the study.

5. Did the questionnaire undergo any changes in its questions post the validation and reliability test? Were any questions removed or added? Please provide clarification.

Conclusion

-What are the strengths and limitations of the study?

Conclusion

What are your suggestion for future studies?

Best regards

Reviewer #4: any restrictions on data sharing should be specified.

It is also suggested that the article elaborate more on the social and economic impacts of COVID-19 on mental health.

more detail could be provided on sample selection to address potential biases due to non-probabilistic sampling.

Please provide more information about handling missing or outlier data to increase accuracy.

The manuscript is generally clear and written in standard English. Minor typographical and grammatical errors exist (e.g., “stress and depression of volunteers aiming” could be clearer), and some sections (e.g., the Methods) could be streamlined for conciseness. A proofreading step before submission is recommended.

Additional Comments for Improvement:

Reorganize the Results section to highlight key findings concisely, avoiding redundancy with tables.

Expand the discussion on how the findings inform future disaster recovery policies, specifically within urban contexts similar to Duran.

Provide clearer links between psychosocial findings and broader systemic risks like climate change.

A broader description of forward-looking policies: The discussion could elaborate more on how the findings could be implemented in future disaster management and reconstruction plans, especially in cities with similar vulnerability structures.

Emphasis on practical solutions: The discussion could focus more on practical solutions and policy suggestions for reducing stress and depression in vulnerable populations.

Reviewer #5: Dear Authors

Thank you for your well performed study. The research is methodologically sound, and the integration of psychosocial health into disaster risk management is a commendable approach that can fill a significant gap in disaster preparedness literature. The findings related to gender, housing conditions, and chronic illnesses offer a critical perspective on psychosocial vulnerability in disaster contexts.

Reviewer #6: Introduction

• Line 68-181: While the introduction contains important evidence about COVID-19, psycho-social health, it also includes excessive descriptions that make it less focused. Some parts of the introduction appear unrelated to the research topic. For example, the integration of psychosocial health into disaster risk management is inadequately described and lacks sufficient context about Ecuador’s specific situation. I would suggest streamlining this section and ensure that all the introduction content directly supports the study’s objectives.

Methodology

• Line 183: The calculation of the sample size, the sampling distribution, and sampling representativeness of the study population are not clearly outlined in the methodology.

• Line 188-194: The study population under the methodology part describes about collection of blood samples and the use of non-probability sampling to select participants, however it does not adequately introduce the study area and population. I would suggest a clearer explanation of where the study was conducted and the characteristics of the study population.

• Line 196 – 213: The term “survey” is not clearly defined in the methodology. The description appears to mix different components of the methods part, including data collection, data collection tools, and other aspects. This makes it difficult for readers to determine which aspect of the methodology is being discussed.

• Line 234: The statistical categorization and analysis used for the modified Spanish version of the Perceived Stress Scale (PSS) and the Patient Health Questionnaire-9 (PHQ-9) need to be explicitly detailed and a clearer explanation of how these tools were analyzed would strengthen the methodological rigor of the study.

Results

• Line 257: The process by which the 340 participants were selected to complete the survey is not clearly described.

• Line 265: The terminology used to describe educational levels, such as “some level of high school education,” “some level of elementary school,” and “some level of undergraduate studies,” appears subjective and lacks precision.

• Line 269: Some variables presented in the table, such as education and marital status, sum precisely to 100%. The table should be checked for consistency to ensure accuracy.

• Line 342: The results section includes a discussion of self-reported stress, which is valuable. However, the methods section should explicitly describe under the methodology part how self-reported data were analyzed.

• Line 376: To improve the identification of risk factors and the prediction of stress levels, I would recommended to include adjusted odds ratio (AOR) results in addition to crude odds ratio (COR) results. This would provide a more comprehensive understanding of the associations between variables after adjusting for potential confounding variables.

Reviewer #7: The study addresses an important and timely issue by investigating the intersection of psychosocial health and disaster risk management in the context of the COVID-19 pandemic. The research is relevant to public health, urban vulnerability, and mental health resilience. However, several key areas require clarification and revision.

The manuscript states that a non-probabilistic sampling strategy was used. However, it does not explain how this might have affected the generalizability of the results. The implications of this method for bias and generalizability were not found. The authors should justify why this method was chosen and discuss its limitations. The recruitment process lacks details about how participants were approached and selected, which is critical for assessing potential biases in the sample. Explain how the sample represents the broader population of Duran. Also, provide evidence/references that the Perceived Stress Scale (PSS) are validated/previously used for the Ecuadorian population?

The manuscript does not state whether potential confounders were adjusted for in the logistic regression models.

Given that multiple factors influence mental health (e.g., socioeconomic status, education, pre-existing conditions, vaccination status), were adjustments made? Provide goodness-of-fit tests and discuss whether multicollinearity was assessed. The study finds that women are more likely to experience depression. However, other factors (such as housing conditions, chronic illnesses, or socioeconomic status) could also contribute to depression.

Causal Language Should Be Avoided. Phrases such as “vaccination acted as a protective factor against depression” should be reworded to reflect association rather than causation. Provide a deeper comparison with global and regional studies, particularly in Latin America.

Reviewer #8: Framework, title, format:

1. The UNDRR has adopted the concept of "Disaster" to include pandemics and infectious diseases rather than focusing solely on natural disasters. The Sendai Framework for Disaster Risk Reduction, established in 2015, reflects discussions from the Convention on the Rights of Persons with Disabilities and emphasizes the growing collaboration between the health and welfare sectors and disaster sectors. The boundary between disasters and pandemics has virtually disappeared. However, why should we revisit the discussion in this manuscript about integrating psychosocial health with disaster risk management strategies?

2. Although this study is titled “Integrating Psychosocial Health into Disaster Risk Management,” it primarily focuses on examining the mental health effects of the COVID-19 pandemic. There is minimal mention of the connection between theories and practices of disaster risk reduction. Therefore, the title should be revised or the argument adjusted to better align with the title.

3. It does not consistently adhere to the specified styles and formatting. This includes the order of manuscript organization, abstract, numbering, and font size of sections and sub-sections, as well as citing references.

- Abstracts should be limited to 300 words and are usually unstructured.

- Cite references using square brackets instead of parentheses (for example, “[1]”).

- Include page numbers.

- These are just a few examples; the document needs significant revision.

Introduction:

4. Throughout history, humans have faced numerous pandemics, including HIV, the Hong Kong flu (H3N2 influenza A virus), SARS, and MERS since the late 20th century. Shouldn't we consider the effects of these past pandemics on psychosocial health in different regions during the discussion of this study?

5. It is crucial to emphasize the importance of the implementation in Ecuador, the universal nature of the results, their general and specific aspects, and the regional context surrounding these outcomes. As you know, people's health and behavior are closely connected to the target region's culture, customs, history, politics, socioeconomic background, and social institutions. Therefore, comparing one country to another or applying findings from one context to a different one may not be suitable. The discussion should differentiate between the regional and universal aspects of the results and then clarify the significance of what this study has uncovered. This also pertains to the novelty and originality of the study.

Methodology, Study area, Results:

6. Figure 1 must be adequately explained in the study area sub-section. This explanation should cover the location, population, urban environment, and social and family systems, among other things. Additionally, it should convey to the reader why this area is most suitable for this study. Regardless, the absence of text in the provided subsection is a significant issue. (Line 183-187)

7. Figures should be appropriately cited and described in the text. (Figure 1 and 3)

8. Multiple surveys are presented, creating the impression of a composite of several studies. Why are different surveys necessary, and what is their connection? For instance, a sub-section on blood sample collection before the "2.1 Survey" sub-section, without any reference to the relationship or necessity between the two, contributes to confusion. Sometimes, it may be necessary to divide the paper into several parts.

9. The section titles and setting sections require revisions to improve their suitability. For example, the sub-sections "2.1 Survey,” "2.2 Perceived Stress Scale," and “2.3 Patient Health Questionnaire-9 for Depression” should not maintain a parallel structure.

10. It is unclear which surveys and analysis methods are associated with which results.

Results:

11. What does this research reveal as the most significant new finding? Aren't all the at-risk individuals mentioned in the results those who have long been considered vulnerable in many studies? The abstract highlights that women experienced more significant psychosocial impacts; however, are these results different from those of the other countries mentioned in the introduction?

Discussion:

12. In the Discussion, it is essential to provide a scientifically valid interpretation based on the statistical trends and correlations found in the Results. This should include insights from previous studies in epidemiology, social sciences, and the humanities (these studies should be referenced in the introduction) and information about the study area.

13. It is essential to emphasize pre-disaster preparedness, as indicated by the term DRR (Disaster Risk Reduction) in the SFDRR. A categorized list of the study's findings and identified countermeasures, organized by disaster phase and target attribute, would enhance understanding.

14. How can perceptions of stress be lower among those experiencing high stress levels? At a minimum, what do the results of this study reveal about the challenges, blind spots, and bottlenecks in policies and efforts during the COVID-19 pandemic?

15. If the results indicate that married women are at a significantly higher risk than others, it is crucial to explore which policies and support measures could effectively address their needs based on the findings. What should the family, community, religion, and government have done?

16. Why are participants living in areas without paved roads experiencing higher stress levels (due to social and environmental factors)? Is it reasonable to classify residents of areas without paved roads as informal settlers? What is the reasoning behind this?

17. The reasons for elevated stress levels among those living in rental apartments require a more thorough explanation. What types of living environments, price ranges, and occupations do rental apartments in the target area typically accommodate, and how do these compare to other housing options?

18. A sub-section titled "Moving forward" should not be included as a subsection at the beginning of the Discussion. Instead, it should be placed at the end of the conclusion, depending on the content of the Discussion, if you wish to include it.

Reviewer #9: The manuscript presents a timely and relevant discussion on the integration of psychosocial health into disaster risk management, drawing insights from the COVID-19 experience. The study provides valuable perspectives on the intersection of mental health and disaster preparedness, which is crucial for strengthening resilience in future public health crises. The manuscript is well-structured, but certain areas require further clarification and enhancement. To enhance the manuscript, read and cite the following related articles in the discussion: https://doi.org/10.3390/ijerph18115719, https://doi.org/10.3389/fpsyt.2021.656664, https://doi.org/10.1017/dmp.2021.140, https://doi.org/10.1371/journal.pone.0280144, https://doi.org/10.1080/13623699.2021.1950519, and https://www.thelancet.com/journals/lanpsy/article/PIIS2215-0366(24)00354-7/abstract

6. PLOS authors have the option to publish the peer review history of their article (what does this mean? ). If published, this will include your full peer review and any attached files.). If published, this will include your full peer review and any attached files.

**Do you want your identity to be public for this peer review?** For information about this choice, including consent withdrawal, please see our For information about this choice, including consent withdrawal, please see our Privacy Policy ..

Reviewer #1: No

Reviewer #2: No

Reviewer #3: No

Reviewer #4: No

Reviewer #5: **Yes:** Dr Reza HabibisaraviDr Reza Habibisaravi

Reviewer #6: **Yes:** Abdulnasir AbageroAbdulnasir Abagero

Reviewer #7: No

Reviewer #8: No

Reviewer #9: No

While revising your submission, please upload your figure files to the Preflight Analysis and Conversion Engine (PACE) digital diagnostic tool, https://pacev2.apexcovantage.com/ . PACE helps ensure that figures meet PLOS requirements. To use PACE, you must first register as a user. Registration is free. Then, login and navigate to the UPLOAD tab, where you will find detailed instructions on how to use the tool. If you encounter any issues or have any questions when using PACE, please email PLOS at . PACE helps ensure that figures meet PLOS requirements. To use PACE, you must first register as a user. Registration is free. Then, login and navigate to the UPLOAD tab, where you will find detailed instructions on how to use the tool. If you encounter any issues or have any questions when using PACE, please email PLOS at figures@plos.org . Please note that Supporting Information files do not need this step.. Please note that Supporting Information files do not need this step.

---

## [Author Response · Author response to Decision Letter 1]

1 Oct 2025

Dear Editor

Title: Integrating Psychosocial Health into Disaster Risk Management: Insights from COVID-19 in Duran, Ecuador.

The authors would like to thank the Editor and Reviewers for their suggestions, which have helped us to improve the paper significantly. As you will see, we have addressed all the comments provided and produced an improved manuscript.

In this document, we have added our response for each comment as an indented text to organize the discussion of the material in this reply. We thank the reviewers for their valuable feedback, constructive comments, and suggestions.

Reviewer # 1

Comment / Answer to comment / Lines/Pages in manuscript

1. Subheading 3.1 - The statistical methods should be explained with procedures, interpretations, and equations/formulas.

This has been added.

In the statistical analysis section, we added a complete explanation of the correlation and regression analysis with the formula required. We eliminated numbering headings.

Lines 256-294.

2. Discuss these articles which focus on disaster vulnerability.

https://link.springer.com/article/10.1007/s10668-023-04149-1

https://link.springer.com/article/10.1007/s00477-022-02267-2

Thank you for the references on disaster vulnerability factors, we have included them in the Introduction section

Reference 37 and 38

Lines 119 -121 - 515

3. The subheading numbering should be properly arranged.

We have rearranged the headings, subheadings that were edited to adhere to the journal guidelines. We eliminated the numbering.

In the whole document.

4. The results should be justified with the support of relevant published articles.

Thank you for your comment. The interpretation of the results with relevant published articles were included in the discussion.

Lines 432 -517

5. The influence of all variables (mentioned in the Tables) on disaster vulnerability should be explained in the Methods section.

The Methods/Survey is under the subsection "Vulnerability Framework Variables", it explains the vulnerability framework, and the variables that are related to the impacts of mental health.

Lines 203 - 216

6. Improve the quality of English to rectify the typo and grammatical errors.

The manuscript has been revised and corrected for typos and grammar.

In the whole document.

7. Keywords - Include only 5 keywords which are relevant.

The manuscript now has just 5 keywords

Lines 65 - 66

Reviewer # 2

Comment / Answer to comment /Lines/Pages in manuscript

1. Photos are not clear enough

Diagrams were converted to TIIF format to improve the resolution and integration in the word document.

Attach as tiff files

2. Please give a brief description of the study area Duran such as population, economy, orientation, etc. to help the reader understand it

We have added a concise description of Duran's main characteristics in the Study Area subsection.

Lines 145-153

3. The introduction is too long and not very relevant.

Thank you for your comment to improve our Introduction: We have restructured the Introduction to incorporate the most relevant concepts related to psychosocial health and Disaster Risk Management (DRM), highlight the research problem concerning the lack of integration between these two fields, and explain why this integration is particularly important in our region. We also emphasize the usefulness of this type of research for decision-makers and strategy implementation. We hope that the revised Introduction will better guide the reader through the rest of the paper.

Lines 67 -142

4. The methodological content is too simple and some necessary methodologies are not considered.

The methodological content has been expanded in the Methods section, including statistical analysis to clarify the methodological process.

Lines 144 - 244

5. 221-223 grading criteria please add references.

The scoring has been described in each Measure (PSS-10 and PHQ-9) subheading. We have added the References: Ruisoto P, López-Guerra VM, Paladines MB, Vaca SL, Cacho R. Psychometric properties of the three versions of the Perceived Stress Scale in Ecuador. 2020; Available from: https://doi.org/10.1016/j.physbeh.2020.113045

Carroll HA, Hook K, Perez OFR, Denckla C, Vince CC, Ghebrehiwet S, et al. Establishing Reliability and Validity for Mental Health Screening Instruments in Resource-Constrained Settings: Systematic Review of the PHQ-9 and Key Recommendations. Psychiatry Res. 2020;291:113236. doi: 10.1016/j.psychres.2020.113236

Ref 54, 57

Lines 228-254

6. P-value should be italicized.

Thank you. This has been updated in the tables

Tables 2,3,4

7. How was the independent variable selected?

The independent variables were selected based on the literature on psychosocial vulnerability and adaptive capacity in the context of disasters and health crises. Specifically, individual and urban sensitivity variables identified in our previous studies (references 2, 3, 26, 29, 31, 43 of the manuscript). See Vulnerability Framework subheading

Lines 202 -225

8. Is OR corrected and how are confounders considered?

We thank you for your thoughtful contribution. To control for confounding, we employed a two-stage approach. First, as a precursor to model selection, we tested how strong the association between stress and depression level and a particular predictor, employing appropriate measures of association between categorical data —Kendall’s Tau-b, Tau-c, and Cramér’s V (Table 2). With these coefficients, we selected which predictors to include in the model. Second, because stress and depression levels had many categories, we employed a multinomial logistic regression model. Employing this methodology, we estimated adjusted odds ratios for various covariates. Comparing crude and adjusted ORs, we identified which had confounding effects—those that altered association magnitude or direction. This modeling technique confirmed that observed associations reflect independent effects and controlled for confounding impact of other applicable covariates. Multinomial logistic regression provided a statistically rigorous basis for the investigation of the contribution of a particular predictor to stress and depression levels. See subheading Regression analysis

Lines 273 - 274

9. How was sample size calculation considered?

In non-probabilistic sampling, it is not appropriate to apply measures designed for probability-based sampling to assess representativeness (Baker et al. 2013). The non-probabilistic method used in Duran’s study was based in the representativeness of system (Duran city), by selecting sites based on in-depth knowledge of Duran’s urban structure and social vulnerability (Borbor-Cordova et al., 2020). Additionally, the use of the multinomial logistic regression model enhances the robustness of our findings, considering that it is suitable for analyzing complex relationships among multiple variables (vulnerability variables), even with non-probabilistic sampling. We argue that the deliberate selection of sites—based on contextual knowledge of urban structure, social vulnerability, and —was essential in mitigating bias during survey administration. See subheading Study design and settings.

Baker R, Brick JM, Bates NA, Battaglia M, Couper MP, Dever JA, et al. Summary report of the AAPOR task force on non-probability sampling. J Surv Stat Methodol. 2013 Nov 1;1(2):90–105

Lines 155 -175

Reviewer #3

Comment / Answer to comment / Lines/Pages in manuscript

Introduction

Please bring the following items:

- Definition of the research problem

- The magnitude and importance of the study variable

- Expressing the necessity of conducting the study

- Finally, the practical purpose of the study should be stated.

We have restructured the Introduction to include the most relevant concepts related to Psychosocial health and Disaster Risk Management, the research problem related to the lack of integration of these approaches, why is relevant in the context of cities and the usability this kind of research for decision makers and implementing strategies at city level. We hope this Introduction will guide the reader into the paper.

Lines 67 - 78

Lines 91 -100

Lines 115-121

Method

1. The method section should include the sample size formula.

In non-probabilistic sampling, it is not appropriate to apply measures designed for probability-based sampling to assess representativeness (Baker et al. 2013). The non-probabilistic method used in Duran’s study was based in the representativeness of system (Duran city), by selecting sites based on in-depth knowledge of Duran’s urban structure and social vulnerability (Borbor-Cordova et al., 2020). Additionally, the use of the multinomial logistic regression model enhances the robustness of our findings, considering that it is suitable for analyzing complex relationships among multiple variables (vulnerability variables), even with non-probabilistic sampling. We argue that the deliberate selection of sites—based on contextual knowledge of urban structure, social vulnerability, and —was essential in mitigating bias during survey administration. See Study design and settings.

Baker R, Brick JM, Bates NA, Battaglia M, Couper MP, Dever JA, et al. Summary report of the AAPOR task force on non-probability sampling. J Surv Stat Methodol. 2013 Nov 1;1(2):90–105

Lines 155 -175

2. Exclusion and inclusion criteria are fully stated.

The inclusion and exclusion criteria are stated in the subsection "Study design and settings."

Lines 159-161.

3. The author did not mention any incomplete, incorrect or corrupt questionnaire

Out of the 446 questionnaires, we excluded those that did not have completed the questionnaires on stress and depression, ending with 340. See subsection "Study design and settings" for description

Lines 171-175

4. The first paragraph of the discussion should be the overall result of the study.

Thank you for your comment, the first paragraph states the overall results of the study.

Lines 432 - 441

5. Did the questionnaire undergo any changes in its questions post the validation and reliability test? Were any questions removed or added? Please provide clarification.

The questionnaire remained unchanged after the validation and reliability tests, with no items added or removed. The final version used in the study was fully consistent with the validated instrument. Validation details are included in the supplementary material (PSS-6 and PHQ-9) for transparency. See section Measures.

Line 227

Conclusion

1. What are the strengths and limitations of the study?

Thank you for your comments, we have added the strengths and explain the limitations of the study. See the Strengths and Limitations section

Lines 576 - 588

2. What are your suggestion for future studies?

We have included them in section 'Conclusions and future work'.

Lines 590 – 621

Reviewer #4

Comment / Answer to comment /Lines/Pages in manuscript

1. Any restrictions on data sharing should be specified.

We appreciate the reviewer’s attention to data sharing compliance. In alignment with Ecuador’s Organic Law on Data Protection (Ley Orgánica de Protección de Datos Personales, 2021), we have ensured that our data handling practices adhere to national regulations and the approval of the ethics committee. Thus, we have prepared an anonymized dataset that complies with national data protection standards and is now available for public sharing.

DOI: 10.6084/m9.figshare.28784126

2. It is also suggested that the article elaborate more on the social and economic impacts of COVID-19 on mental health.

In the subsections of the Discussion, we have elaborated more on the social and economic impacts of COVID-19 on mental health, supported by references from global and local examples.

Lines 443 - 517

3. More detail could be provided on sample selection to address potential biases due to non-probabilistic sampling.

This has been added in subheading Study design and settings.

We would like to provide clarification regarding our sampling methodology: due to logistical constraints imposed by the COVID-19 pandemic, we employed a non-probabilistic convenience sampling approach for this study. To mitigate potential biases associated with this method, we conducted outreach campaigns across various urban sectors, aiming to maximize the diversity of our sample. Furthermore, we compared the sociodemographic characteristics of our participants with official population data from Durán to assess partial representativeness.

While we acknowledge that non-probabilistic sampling limits the generalizability of our findings, such approaches are often necessary in exploratory research, especially under challenging circumstances like a pandemic. This methodology allowed us to gain a better understanding of the Duran vulnerability to disaster, which can inform future, more comprehensive studies.

Lines 155 - 175

4. Please provide more information about handling missing or outlier data to increase accuracy.

This has been included.

Missing values in our dataset were minimal. The variable with the most missing values had only 12 out of 340 cases (<3.5%). Due to the low proportion of missing data and following standard practice, imputation was not performed as it would have had minimal impact on the estimates.

We performed the multinomial logistic regression using listwise deletion. This did not affect the significance or interpretation of the results, and parameter estimates remained stable. We, therefore, concluded that missing data did not introduce bias or compromise the robustness of the findings.

No extreme outliers were identified in key variables. Visual inspection and statistical checks confirmed the appropriateness of data distributions before model fitting.

5. The manuscript is generally clear and written in standard English. Minor typographical and grammatical errors exist (e.g., “stress and depression of volunteers aiming” could be clearer), and some sections (e.g., the Methods) could be streamlined for conciseness. A proofreading step before submission is recommended.

This has been resolved. We have revised the whole manuscript for typos and grammar, and we have focused on clarity and conciseness.

All the manuscript

6. Reorganize the Results section to highlight key findings concisely, avoiding redundancy with tables.

Results have been reorganized to highlight the key findings, in subsections of stress and depression.

Lines 349 - 421

7. Expand the discussion on how the findings inform future disaster recovery policies, specifically within urban contexts similar to Duran

In the Discussion subsections we have addressed how the findings inform future recovery by building resilient communities and fostering integrated policy within the urban context using the case of Duran.

Lines 519 - 553

8. Provide clearer links between psychosocial findings and broader systemic risks like climate change.

The section titled "Psychosocial factors and systemic vulnerability" was restructured in the “Urban vulnerability” in which the systemic risk has been addressed.

Lines 502 - 517

9. A broader description of forward-looking policies: The discussion could elaborate more on how the findings could be implemented in future disaster management and reconstruction plans, especially in cities with similar vulnerability structures.

Thank you for your recommendations that has been included in some subsections of the Discussion, Conclusions and Future work.

Lines 590 - 621

10. Emphasis on practical solutions: The discussion could focus more on practical solutions and policy suggestions for reducing stress and depression in vulnerable populations.

This has been included in Discussion subsections of A Pathway to Building Resilient Communities and Integrating Health Care, Mental Health Systems, and Disaster Risk Management

Lines 519 – 574

Reviewer #5:

Dear Authors

Thank you for your well performed study. The research is methodologically sound, and the integration of psychosocial health into disaster risk management is a commendable approach that can fill a significant gap in disaster preparedness literature. The findings related to gender, housing conditions, and chronic illnesses offer a critical perspective on psychosocial vulnerability in disaster cont

---

## [Decision Letter · Decision Letter 1]

8 Dec 2025

PONE-D-24-34167R1Integrating Psychosocial Health into Disaster Risk Management: Insights from COVID-19 in Duran, Ecuador.PLOS One

Dear Dr. Borbor-Cordova,

Thank you for submitting your manuscript to PLOS ONE. After careful consideration, we feel that it has merit but does not fully meet PLOS ONE’s publication criteria as it currently stands. Therefore, we invite you to submit a revised version of the manuscript that addresses the points raised during the review process.

Academic Editor comment: In my opinion authors responded satisfactorily to most of reviewers' comments. However, some minor adjustments are still needed. Please address the new reviewers' comments below and resubmit.

We look forward to receiving your revised manuscript.

Kind regards,

Massimo Palme

Academic Editor

PLOS One

Journal Requirements:

Reviewers' comments:

Reviewer's Responses to Questions

**Comments to the Author**

1. If the authors have adequately addressed your comments raised in a previous round of review and you feel that this manuscript is now acceptable for publication, you may indicate that here to bypass the “Comments to the Author” section, enter your conflict of interest statement in the “Confidential to Editor” section, and submit your "Accept" recommendation.

Reviewer #4: All comments have been addressed

Reviewer #5: (No Response)

Reviewer #6: (No Response)

2. Is the manuscript technically sound, and do the data support the conclusions?

Reviewer #4: Yes

Reviewer #5: Partly

Reviewer #6: Partly

3. Has the statistical analysis been performed appropriately and rigorously? 

Reviewer #4: Yes

Reviewer #5: Yes

Reviewer #6: Yes

4. Have the authors made all data underlying the findings in their manuscript fully available?

Reviewer #4: Yes

Reviewer #5: Yes

Reviewer #6: Yes

5. Is the manuscript presented in an intelligible fashion and written in standard English?

Reviewer #4: Yes

Reviewer #5: Yes

Reviewer #6: Yes

6. Review Comments to the Author

Reviewer #4: (No Response)

Reviewer #5: Dear Author

Thank you for your submission. I read your manuscript and dropped some comments as following:

Introduction

The introduction has been substantially revised through the peer review process and now effectively establishes the research problem. It clearly articulates the gap between current DRM practices and psychosocial health integration, particularly in Global South contexts. However, I think there are some issues that need further attention, as follows:

- Some remaining redundancy (psychosocial health concept explained multiple times)

- Connection between climate hazards and COVID-19 could be more explicit

- Limited discussion of why Durán specifically is relevant beyond prior climate studies

- Some references dated (e.g., reference to 2010 census data for Durán population)

- Could include more recent DRM integration literature post-2020 ( also in discussion)

- Missing some recent systematic reviews on pandemic mental health ( also in discussion)

Methodology

Methodology is clear and reproducible, though the non-probabilistic sampling approach warrants careful consideration for interpreting results. There are some comments in this section to help for stronger methods as following:

- Please describe sample size calculation with justification

- Limited discussion of how multicollinearity among vulnerability variables was addressed

-There is no mention of data validation procedures

Results

Results are presented accurately and clearly with appropriate use of tables and figures. Data presentation is generally well-organized. There are just some comments to help you for better result presentation as following:

- Tables 3 & 4 are dense; some extreme OR values (>1000) indicate complete separation in logistic regression, limiting interpretability.

- Some confidence intervals extremely wide, suggesting sparse data in certain subgroups.

Discussion

The discussion is comprehensive, well-structured, and effectively interprets findings in context of existing literature. Recent revisions have significantly improved regional contextualization. There are more issues to consider:

- Limited discussion of potential reverse causality (does depression predict lower vaccination rates or vice versa?)

- Could discuss intersectionality more explicitly (overlapping gender, age, economic vulnerabilities)

- Implications for disaster preparedness could be more concrete

- Limited discussion of cultural adaptations needed for interventions in Durán context

Conclusions

Well written. Conclusion is evidence-based and consistent with research objectives, appropriately synthesizing findings and providing actionable recommendations.

References and Citations

References are properly formatted and citations are comprehensive, though formatting consistency has been improved.

Some older citations (2010 census data for Durán) could be updated with more recent statistics if available. There are more comments:

- Missing some recent COVID-19 mental health systematic reviews (2021-2024)

- Limited representation of Global South-specific disaster resilience literature

Reviewer #6: Line 154: The study was conducted in 2021; therefore, some of the findings and a portion of the referenced literature may now be considered outdated. Updating these elements could strengthen the manuscript.

Line 154: I continue to have reservations regarding the representativeness of the non-probabilistic convenience sampling approach used for the quantitative survey. Clarification or justification of this method would be helpful.

Line 164: The rationale for selecting community centers as recruitment sites appears somewhat subjective. Further explanation on how logistical feasibility, infrastructure adequacy, and safety considerations informed this decision would enhance clarity.

Line 168: It is unclear why 106 out of the 446 individuals who initially participated did not complete the study and were subsequently excluded. Providing additional detail on the reasons for non-completion would strengthen the methodology section.

Line 176: The explanation that sites were selected to capture urban neighborhood diversity and based on logistical feasibility may still appear subjective. Additional justification on how these criteria were applied would be beneficial.

Line 183: Please clarify the type of consent obtained and how it was documented electronically to ensure transparency and adherence to ethical standards.

7. PLOS authors have the option to publish the peer review history of their article (what does this mean? ). If published, this will include your full peer review and any attached files.). If published, this will include your full peer review and any attached files.

**Do you want your identity to be public for this peer review?** For information about this choice, including consent withdrawal, please see our For information about this choice, including consent withdrawal, please see our Privacy Policy ..

Reviewer #4: No

Reviewer #5: **Yes:** Reza Habibisaravi, MD PhDReza Habibisaravi, MD PhD

Faculty Member

School of Allied Medical Sciences

Mazandaran University of Medical Sciences, Sari, Iran

Reviewer #6: No

---

## [Author Response · Author response to Decision Letter 2]

29 Jan 2026

PONE-D-24-34167R1

Integrating Psychosocial Health into Disaster Risk Management: Insights from COVID-19 in Duran, Ecuador.

Below is our detailed response to the reviewer comments, with associated line numbers in the updated manuscript. We look forward to your review and response.

Comments to the Author

1. If the authors have adequately addressed your comments raised in a previous round of review and you feel that this manuscript is now acceptable for publication, you may indicate that here to bypass the “Comments to the Author” section, enter your conflict of interest statement in the “Confidential to Editor” section, and submit your "Accept" recommendation.

Reviewer #4: All comments have been addressed

Reviewer #5: (No Response)

Reviewer #6: (No Response)

2. Is the manuscript technically sound, and do the data support the conclusions?

Reviewer #4: Yes

Reviewer #5: Partly

Reviewer #6: Partly

3. Has the statistical analysis been performed appropriately and rigorously?

Reviewer #4: Yes

Reviewer #5: Yes

Reviewer #6: Yes

4. Have the authors made all data underlying the findings in their manuscript fully available?

Reviewer #4: Yes

Reviewer #5: Yes

Reviewer #6: Yes

5. Is the manuscript presented in an intelligible fashion and written in standard English?

Reviewer #4: Yes

Reviewer #5: Yes

Reviewer #6: Yes

6. Review Comments to the Author

Reviewer #4: (No Response)

Reviewer #5: Dear Author

Thank you for your submission. I read your manuscript and dropped some comments as following:

Introduction

The introduction has been substantially revised through the peer review process and now effectively establishes the research problem. It clearly articulates the gap between current DRM practices and psychosocial health integration, particularly in Global South contexts. However, I think there are some issues that need further attention, as follows:

- Some remaining redundancy (psychosocial health concept explained multiple times)

Thank you for this point. We removed lines 137-143 to avoid redundancy.

- Connection between climate hazards and COVID-19 could be more explicit

We added a paragraph including specific effects of compound climatic and pandemic risks globally and in our Latin American region.

Lines 61-65: “In line with the Sendai Framework for disaster risk reduction (DRR), humanitarian emergencies, climate and health crises, and compound risks offer critical opportunities to advance resilience in urban and health systems by addressing structural vulnerabilities, enhancing interinstitutional coordination, and integrating mental and psychosocial health into disaster risk reduction and management strategies.”

We also added updated references: Ford et al., 2021, Hartginger et al, 2023; Zavaleta et al., 2021, Uche Ezeh et al., 2021, Borg et al., 2021 and Palmeiro-Silva et al, 2022 on lines 94, 99 111, 104, 113, 491

- Limited discussion of why Durán specifically is relevant beyond prior climate studies

We would like to remark that the city of Durán has functioned as an urban laboratory, since 2018, for examining the intersection of climate hazards, social vulnerability, and COVID-19 through sustained community-based participatory research conducted by our team. This long-term engagement with the local government and communities has enabled a better understanding of the complex interactions among compound urban risks and contributes original empirical evidence to a field where research on intermediate cities in Latin America remains limited.

Line 132-136:“These efforts position the city as an urban laboratory for examining the complex interactions among climate risks, pandemics and social vulnerability in intermediate Latin America cities. Considering the paucity of original research and context-specific evidence, this study offers insights into how vulnerability factors interact during crises, informing strategies to strengthen urban resilience [29,34].”

- Some references dated (e.g., reference to 2010 census data for Durán population)

Line 158: The population of Duran has been updated to 303,910 residents, using data from the 2022 Ecuador’s Census.

- Could include more recent DRM integration literature post-2020 ( also in discussion)

This work also incorporates recent disaster risk management (DRM) and climate adaptation literature that integrates climate change with physical and mental health perspectives, strengthening the analysis of systemic and compounding risks in urban contexts. We also added references as UNDRR&WHO, 2023 on line 72 and Gray et al., 2021 on line 104.

- Missing some recent systematic reviews on pandemic mental health (also in discussion)

We have added several reviews on mental health:

Line 89: Zhang et al, 2022

Line 547: Duden et al, 2022

Line 481: Jia et al, 2024

Lines 104, 523: Gray et al, 2021

Lines 80, 99, 104, 109, 111: Lawrance et al, 2022

Methodology

Methodology is clear and reproducible, though the non-probabilistic sampling approach warrants careful consideration for interpreting results. There are some comments in this section to help for stronger methods as following:

- Please describe sample size calculation with justification

Sample size:

We thank the reviewer for this comment. The sample size was determined based on the logistic and operational capacity of the field campaigns, consistent with the non-probabilistic convenience sampling strategy employed. In this context, the sample size was not based on a formal a priori calculation, as is typical in probabilistic study designs. Nevertheless, with the achieved sample size (n = 340), the study had adequate statistical power (>80%) to detect associations of moderate magnitude (odds ratios approximately between 1.5 and 2.0) at a 5% significance level for the main dichotomous outcomes, contributing to the stability of the estimated logistic regression models. The “Strengths and Limitations” section notes that associations of smaller magnitude were estimated with wide confidence intervals and non-significant results, primarily due to the low number of events in some categories, which is expected given the sample size and data distribution.

The sample size was determined based on the logistic and operational capacity of the field campaigns, consistent with the non-probabilistic convenience sampling strategy employed.

Line 182-185:“Thus, sample size was determined based on the operational capacity of the field campaigns, consistent with the non-probabilistic convenience sampling strategy employed. See Figure 1 for description of the study site and sampling locations [59].”

Lines 581-600, in the "Strengths and Limitations” section notes:

Finally, although the sample size obtained provided adequate statistical power to detect associations of moderate magnitude, some associations, particularly those involving less frequent categories, were estimated with wide confidence intervals and non-significant results. This reflects limited precision due to data scarcity rather than the absence of an underlying association, which has been considered in the interpretation of findings.

- Limited discussion of how multicollinearity among vulnerability variables was addressed

Thank you for the questions to help clarify the data processing. Before conducting multivariable analyses, we assessed multicollinearity among independent variables representing individual vulnerability, urban sensitivity, and adaptive capacity. We examined correlations between covariates and the outcomes (stress and depression), as well as inter-covariate associations, using measures appropriate to variable scale (Spearman or Kendall coefficients for ordinal variables and Cramér’s V for nominal variables). Most associations were weak (<0.2), with the exception of sex and occupational status, which showed a moderate association (~0.4). Although individual correlation values are not reported, this information has now been explicitly clarified in the Correlation Analysis section.

We include in the manuscript:

Line 287: “as well as to evaluate potential multicollinearity among vulnerability-related covariates…”

-There is no mention of data validation procedures

The steps described, data validation and quality control procedures were applied prior to analysis. These included verification of data completeness, detection of out-of-range values, and consistency checks between related variables (e.g., demographic and clinical information). Duplicate records were identified and removed using unique participant codes, and the final dataset was anonymized before analysis. These procedures have now been explicitly described and clarified in the Methods section to improve transparency and reproducibility.

On lines 311-314, we have added a concise description of the data validation procedure.

“Prior to analysis, data validation procedures were conducted,including checks for completeness and internal consistency, with duplicate records removed using unique participant identifiers.”

In lines 311-314, we explain the data preprocessing:

“All statistical analyses were conducted using R (version 4.4.2). Data were validated prior to analysis, and assumptions for each statistical test, including variable scale and distribution, were assessed. When assumptions of normality or homoscedasticity were not met, appropriate non-parametric methods were applied. Missing data were not imputed; analyses were restricted to complete cases.”

Results

Results are presented accurately and clearly with appropriate use of tables and figures. Data presentation is generally well-organized. There are just some comments to help you for better result presentation as following:

- Tables 3 & 4 are dense; some extreme OR values (>1000) indicate complete separation in logistic regression, limiting interpretability.

Tables 3 and 4 were revised to improve interpretability by retaining only stable and interpretable odds ratio estimates. Results affected by complete or quasi-complete separation, or presenting extreme confidence intervals, were not reported as odds ratios and this clarification has been explicitly included in the Results section.

Line 380: Table 3 presents the regression model predicting moderate stress; estimates for high stress were not reported due to very low event counts (<10) and limited interpretability.

Line 380: The new Table 3 is included in the manuscript.

Line 408: New table 4 is included in the manuscript.

- Some confidence intervals extremely wide, suggesting sparse data in certain subgroups.

The presence of very wide confidence intervals has been explicitly indicated in the “Strengths and Limitations” section starting on line 581, clarifying that these estimates are primarily driven by sparse data and low event counts in certain subgroups rather than by the absence of an underlying association.

Discussion

The discussion is comprehensive, well-structured, and effectively interprets findings in context of existing literature. Recent revisions have significantly improved regional contextualization. There are more issues to consider:

- Limited discussion of potential reverse causality (does depression predict lower vaccination rates or vice versa?)

Line 482-488: Thank you for this relevant question. We have revised the manuscript to include recent global literature showing mixed findings, with several studies reporting lower COVID-19 vaccination uptake and higher hesitancy among individuals with depression and anxiety. We also incorporated recent evidence from Durán (Orellana-Manzano et al., 2026), which reported high willingness to follow vaccination guidance, reflecting strong trust in local authorities and health providers despite limited knowledge of COVID-19 transmission. Finally, we clarify that the present study is designed to examine associations rather than to establish causal relationships.

- Could discuss intersectionality more explicitly (overlapping gender, age, economic vulnerabilities)

Lines 512-518: Thank you for your comment. Intersectionality is a critical concept in disaster risk reduction, public health, and climate change adaptation in urban contexts. We have added a paragraph highlighting the study contribution to intersectionality research by identifying how multiple, intertwined vulnerability factors shape mental health outcomes, while also acknowledging that power dynamics, violence, and safety conditions are critical determinants that must be considered in the design and implementation of interventions.

“Duran’s study contributes to intersectionality research by moving beyond the individual level to examine how structural and environmental inequities at the urban scale shape vulnerability [112,113]. Showing that gender, age, health status, inadequate urban infrastructure, residence in informal neighborhoods, and access to knowledge and health information interact to produce distinct patterns of vulnerability that help explain mental health disparities during health-related crises. Besides urban power asymmetries, violence, and the availability of safe spaces emerge as critical contextual factors for the design and implementation of effective interventions [114].”

- Implications for disaster preparedness could be more concrete

To elaborate on implications for disaster preparedness, we added lines 711-715 to the following sentence:

Line 710: “Strengthening community social networks at community level, simultaneously builds resilience to both health emergencies and climate hazards [117].”

Line 711-715: “The Entre Nosotras initiative illustrates culturally adapted, scalable strategies for Ecuador and other Latin American cities by shifting from individual to group-based interventions aligned with community preferences, incorporating shared childcare arrangements, scheduling activities during evenings and weekends, and strengthening safe community spaces to support psychosocial wellbeing [118].”

- Limited discussion of cultural adaptations needed for interventions in Durán context

We added specific interventions at the family and community level that we believe are needed in our cultural context.

Lines 535-542: “The strategies may include psychosocial first aid, skill‑building workshops, family communication plans, and preparedness kits (including children’s items), to foster safety, support, and resilience before disasters occur. Strengthening community social networks at community level, simultaneously builds resilience to both health emergencies and climate hazards [117]. The Entre Nosotras initiative illustrates culturally adapted

---

## [Editor Report · Decision Letter 2]

4 Feb 2026

Integrating Psychosocial Health into Disaster Risk Management: Insights from COVID-19 in Duran, Ecuador.

PONE-D-24-34167R2

Dear Dr. Borbor-Cordova,

We’re pleased to inform you that your manuscript has been judged scientifically suitable for publication and will be formally accepted for publication once it meets all outstanding technical requirements.

Kind regards,

Massimo Palme

Academic Editor

PLOS One
---

## [Editor Report · Acceptance letter]

PONE-D-24-34167R2

PLOS One

Dear Dr. Borbor-Cordova,

I'm pleased to inform you that your manuscript has been deemed suitable for publication in PLOS One. Congratulations! Your manuscript is now being handed over to our production team.

Kind regards,

on behalf of

Dr. Massimo Palme

Academic Editor

PLOS One